# Phenylbutyrate rescues the transport defect of the Sec61α mutations V67G and T185A for renin

Mark Sicking[1], Martina Živná[2], Pratiti Bhadra[3], Veronika Barešová[2], Andrea Tirincsi[1] (ID), Drazena Hadzibeganovic[1], Kateřina Hodaňová[2], Petr Vyleťal[2], Jana Sovová[2], Ivana Jedličková[2], Martin Jung[1] (ID), Thomas Bell[4], Volkhard Helms[3] (ID), Anthony J Bleyer[2,5], Stanislav Kmoch[2], Adolfo Cavalié[6], Sven Lang[1] (ID)

**The human Sec61 complex is a widely distributed and abundant molecular machine. It resides in the membrane of the endoplasmic reticulum to channel two types of cargo: protein substrates and calcium ions. The *SEC61A1* gene encodes for the pore-forming Sec61α subunit of the Sec61 complex. Despite their ubiquitous expression, the idiopathic *SEC61A1* missense mutations p.V67G and p.T185A trigger a localized disease pattern diagnosed as autosomal dominant tubulointerstitial kidney disease (ADTKD–*SEC61A1*). Using cellular disease models for ADTKD–*SEC61A1*, we identified an impaired protein transport of the renal secretory protein renin and a reduced abundance of regulatory calcium transporters, including SERCA2. Treatment with the molecular chaperone phenylbutyrate reversed the defective protein transport of renin and the imbalanced calcium homeostasis. Signal peptide substitution experiments pointed at targeting sequences as the cause for the substrate-specific impairment of protein transport in the presence of the V67G or T185A mutations. Similarly, dominant mutations in the signal peptide of renin also cause ADTKD and point to impaired transport of this renal hormone as important pathogenic feature for ADTKD–*SEC61A1* patients as well.**

## Introduction

Since the advent of next generation sequencing (NGS) an ever-growing number of genetic mutations have been identified. Thus, more and more human patients suffering from a disease of unknown etiology receive a diagnosis from such NGS efforts and the affected gene causing or contributing to the development of the patient's illness is identified. Undoubtedly, integration of the NGS technology into clinical practice is one of the key advancements in personalized medicine and multiple successful cases with molecular-guided therapies have been reported, including cases of rare and ultra-rare genetic diseases. Although treatment options for rare genetic disease are still scarce, molecular diagnosis by NGS and personalized medicine also opened up an avenue of low-cost, low-risk approaches such as dietary adjustments or repurposing of clinically approved drugs (Boycott et al, 2013; Lee & Choi, 2016; Liu et al, 2019).

The human *SEC61A1* gene encodes for the pore-forming Sec61α subunit of the heterotrimeric Sec61 complex (Rapoport et al, 2017; Gemmer & Förster, 2020; O'Keefe & High, 2020). This complex provides a hub for both the transport of unfolded polypeptides in and across the membrane of the ER as well as for the efflux of calcium ions ($Ca^{2+}$) from the lumen of the ER (Erdmann et al, 2011; Lang et al, 2011, 2012). The protein and $Ca^{2+}$ permeability of the Sec61 complex can be demonstrated using living cells under native conditions, as well as in in vitro settings based on biochemical purification and reconstitution of the complex into artificial membranes (Görlich & Rapoport, 1993; Wirth et al, 2003; Erdmann et al, 2009; Nguyen et al, 2018). Multiple structural studies highlight the evolutionarily conserved architecture of the Sec61 complex (Van den Berg et al, 2004; Zimmer et al, 2008; Becker et al, 2009; Egea & Stroud, 2010; Gogala et al, 2014; Voorhees et al, 2014; Tanaka et al, 2015). The central, pore-forming Sec61α subunit has ten transmembrane helices (TMHs) and a design reminiscent of an hourglass. Functionally relevant key elements include the central constriction zone called the pore ring that separates the two opposing funnels, the plug domain occupying the luminal funnel in the closed conformation of the Sec61 channel, and the lateral gate formed by TMHs 2 and 7 allowing the lateral release of polypeptides into the membrane (Junne et al, 2006; Lang et al, 2017, 2019).

Recently, five dominant point mutations from patients suffering from different rare diseases were identified individually in the open reading frame of the ubiquitously expressed *SEC61A1* gene (Bolar et al, 2016; Schubert et al, 2017; Van Nieuwenhove et al, 2020). The heterozygous *SEC61A1* alleles identified in humans include four

[1]Department of Medical Biochemistry and Molecular Biology, Saarland University, Homburg, Germany  [2]Research Unit for Rare Diseases, Department of Pediatrics and Metabolic Disorders, First Faculty of Medicine, Charles University, Prague, Czech Republic  [3]Center for Bioinformatics, Saarland University, Saarbrücken, Germany  [4]Department of Chemistry, University of Nevada, Reno, NV, USA  [5]Section on Nephrology, Wake Forest School of Medicine, Winston-Salem, NC, USA  [6]Experimental and Clinical Pharmacology and Toxicology, Pre-clinical Center for Molecular Signaling (PZMS), Saarland University, Homburg, Germany

Correspondence: sven.lang@uni-saarland.de

missense mutations causing the amino acid substitutions V67G (plug domain), V85D (pore ring residue, TMH 2), Q92R (TMH 2), and T185A (TMH 5) as well as one nonsense mutation introducing a premature stop at E381* (TMH 8). The rare-diseases arising from those mutations are (i) common variable immunodeficiency for V85D and E381* (Schubert et al, 2017), (ii) autosomal dominant severe congenital neutropenia for V67G (Bolar et al, 2016) and Q92R (Van Nieuwenhove et al, 2020), and (iii) autosomal dominant tubulointerstitial kidney disease (ADTKD) for V67G and T185A (Bolar et al, 2016; Espino-Hernández et al, 2021).

ADTKD encompasses a group of hereditary chronic kidney diseases characterized by autosomal dominant inheritance, the absence of proteinuria and hematuria, renal tubular and interstitial abnormalities, and the progressive loss of kidney function resulting in end-stage kidney disease between the $3^{rd}$ and $7^{th}$ decade of life. Renal transplantation is the ultimate recommended treatment of the monogenetic disease, as it will not recur in the transplanted kidney (Ayasreh et al, 2017; Bleyer et al, 2017). Since the introduction of the gene-based terminology by KDIGO in 2015, ADTKD has been categorized into currently five subtypes depending on the genetic defect (Eckardt et al, 2015). Other than mutations in the *SEC61A1* gene, mutations in genes encoding for mucin 1, the transcription factor HNF1β, uromodulin, and renin can cause ADTKD (Devuyst et al, 2019). Although the clinical and laboratory features of patients affected by a *SEC61A1*-associated disease are in general well characterized, the detailed molecular mechanisms giving rise to the tissue- and organ-specific defects in spite of the ubiquitous expression of the *SEC61A1* gene are still unclear and a matter of debate.

Here, we identified specific defects associated with the ADTKD–*SEC61A1* variants V67G and T185A at the cellular level. In addition to a reduced abundance of $Ca^{2+}$ transporters affecting $Ca^{2+}$ homeostasis, we identified a substrate-specific impairment of protein transport for the renal secretory protein renin. We managed to rescue the hampered protein transport of renin by a signal peptide exchange, that is, by a genetic approach, and by pharmacological means via repurposing the Food and Drug Administration (FDA) approved small molecule sodium phenylbutyrate (PBA). PBA was originally approved for the treatment of urea cycle disorders. Taken together, our data provide novel insights into the pathogenic mechanisms in the kidney that may very well contribute to the tissue-specific symptomatic of ADTKD–*SEC61A1* patients, which seem to originate in a substrate-specific malfunction of protein transport that further spawn a dysregulation of $Ca^{2+}$ homeostasis.

## Results

### Reduced steady-state levels of calcium flux regulators in ADTKD–*SEC61A1* cells

To address the functional consequences of the ADTKD-associated Sec61α mutations V67G and T185A at the molecular level, we used stably transfected human embryonic kidney (HEK) cell lines. In addition to the endogenous *SEC61A1* gene these stable HEK293 lines harbor a C-terminally FLAG-tagged *SEC61A1* transgene that encodes

either the wild type (WT) or one of the two mutant Sec61α proteins (V67G, T185A). Thus, the cell lines mimic the heterozygous genetics of the ADTKD–*SEC61A1* patients and allow comparison to the WT control. Before performing functional experiments with those cell lines, we analyzed the steady-state levels of various key factors involved in polypeptide transport into and calcium leakage out of the ER.

As summarized in Fig 1, we first determined the relative protein content of the Sec61 complex in V67G and T185A cells compared with the WT control. The α- and β-subunit of the Sec61 complex showed slightly elevated levels in ADTKD–*SEC61A1* cells. Of note, protein levels of Sec61α were determined independently by antibodies recognizing either the FLAG tag or Sec61α itself. The latter also verifies the heterozygous nature of the cell lines as a second band for Sec61α-FLAG is appearing only in WT, V67G, and T185A cells (Fig 1B). Whereas the two bands ran too close for individual quantification, in all three cell lines, the bands for the endogenous and FLAG-tagged Sec61α appear in roughly equimolar intensities. In addition, presence of the single-point mutations causing the amino acid exchanges V67G and T185A was verified by sequencing of genomic DNA fragments (Fig S1A–C). Well-characterized proteins that either support or interact with the Sec61 complex, including subunits of the targeting receptors SRβ, Caml, and hSnd2 (Akopian et al, 2013; Aviram & Schuldiner, 2017; Haβdenteufel et al, 2017), ER luminal chaperones such as BiP, Grp170, and PDI (Behnke et al, 2015; Melnyk et al, 2015), or allosteric effectors and an oligosaccharyltransferase (OST) subunit that are part of the ER protein translocase including Sec62, TRAPα, and Rpn1 (Gemmer & Förster, 2020; Lang et al, 2019) did not show relevant alterations in their steady-state levels (Fig 1). However, when evaluating levels of regulators important in cellular and ER calcium homeostasis, we identified two $Ca^{2+}$ shuttling membrane proteins, Orai1 and SERCA2, with reduced steady-state levels in both types of ADTKD–*SEC61A1* cells. Whereas SERCA2 is a $Ca^{2+}$ ATPase of the ER membrane, Orai1 is a $Ca^{2+}$ channel of the plasma membrane that can be activated by an ER protein such as STIM1 via direct coupling (Clapham, 2007; Periasamy & Kalyanasundaram, 2007; Shaw & Feske, 2012). Measurement of STIM1 protein levels in ADTKD cells revealed a reduction only in the presence of the Sec61α-T185A mutation (Fig 1).

Thus, in assaying the protein abundance landscape of the ADTKD–*SEC61A1* cells, we identified reduced levels of crucial regulators of calcium homeostasis like Orai1 and SERCA2 in the presence of the dominant Sec61α-V67G or T185A mutation.

### Distinct aberrations of the $Ca^{2+}$ homeostasis in ADTKD–*SEC61A1* cells

Given the reduced abundance of the $Ca^{2+}$ handling proteins SERCA2 and Orai1, we first explored mechanisms of cellular $Ca^{2+}$ homeostasis. Using the ratiometric $Ca^{2+}$ indicator Fura-2, sensitive live-cell imaging was performed to measure potentially altered intracellular $Ca^{2+}$ levels in ADTKD–*SEC61A1* cells (Fig 2).

Importantly, the Sec61 complex is a crucial regulator of intracellular $Ca^{2+}$ gradients, acting as major $Ca^{2+}$ leak channel of the ER (Gamayun et al, 2019; Lang et al, 2011). This perpetual leakage of ER $Ca^{2+}$ is counterbalanced by SERCA2 to preserve the steep $Ca^{2+}$ gradient that exists between the ER lumen ($Ca^{2+}$ in μM range) and

**A**

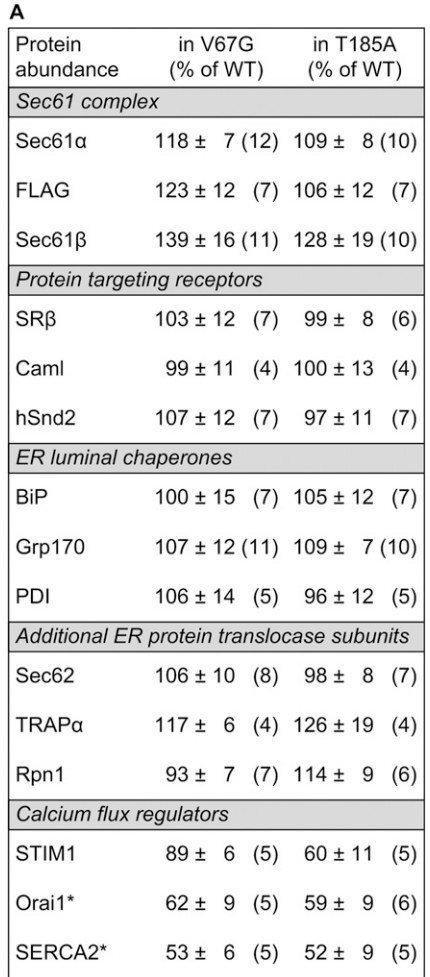

| Protein abundance | in V67G (% of WT) | in T185A (% of WT) |
|---|---|---|
| *Sec61 complex* | | |
| Sec61α | 118 ± 7 (12) | 109 ± 8 (10) |
| FLAG | 123 ± 12 (7) | 106 ± 12 (7) |
| Sec61β | 139 ± 16 (11) | 128 ± 19 (10) |
| *Protein targeting receptors* | | |
| SRβ | 103 ± 12 (7) | 99 ± 8 (6) |
| Caml | 99 ± 11 (4) | 100 ± 13 (4) |
| hSnd2 | 107 ± 12 (7) | 97 ± 11 (7) |
| *ER luminal chaperones* | | |
| BiP | 100 ± 15 (7) | 105 ± 12 (7) |
| Grp170 | 107 ± 12 (11) | 109 ± 7 (10) |
| PDI | 106 ± 14 (5) | 96 ± 12 (5) |
| *Additional ER protein translocase subunits* | | |
| Sec62 | 106 ± 10 (8) | 98 ± 8 (7) |
| TRAPα | 117 ± 6 (4) | 126 ± 19 (4) |
| Rpn1 | 93 ± 7 (7) | 114 ± 9 (6) |
| *Calcium flux regulators* | | |
| STIM1 | 89 ± 6 (5) | 60 ± 11 (5) |
| Orai1* | 62 ± 9 (5) | 59 ± 9 (6) |
| SERCA2* | 53 ± 6 (5) | 52 ± 9 (5) |

**B**

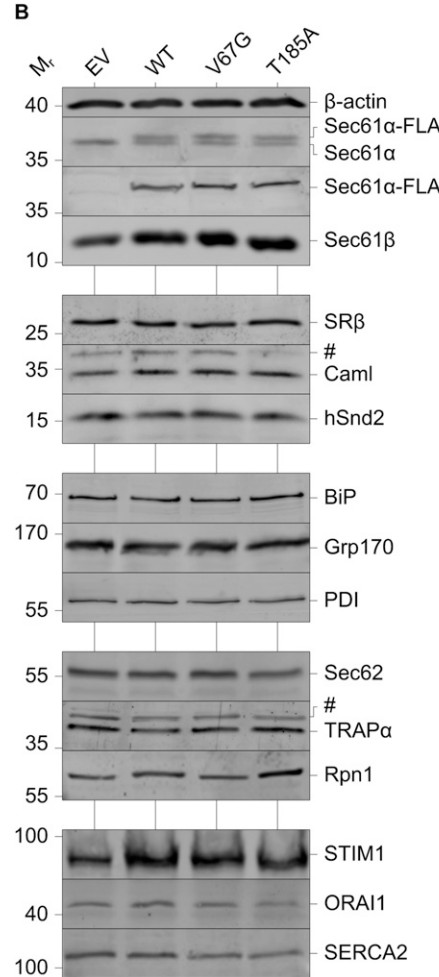

**Figure 1. Comparison of the steady-state protein abundance in the ADTKD–*SEC61A1* cell models.**
**(A)** Protein content was determined by quantification of Western blots for various proteins. A total of 15 proteins plus β-actin as loading reference were analyzed and categorized into five groups based on functionally relevant key terms. The values given represent the mean ± SEM. N as number of blot signals analyzed is given in brackets. Data were normalized to the wild-type (WT) control cells (% of WT). Protein names marked with an asterisk were significantly reduced ($P < 0.05$) in both ADTKD–*SEC61A1* cells, the heterozygous V67G and T185A line.
**(B)** Representative Western blot panels are shown for each of the analyzed proteins. Cross-reactions are marked by the hashtag symbol (#). Note the presence of the additional Sec61α-FLAG band occurring in the heterozygous cell models for WT, V67G, and T185A, but is absent in the empty vector (EV) cells.

the cytosol (Ca²⁺ in nM range) under resting conditions. Based on the reduced levels of SERCA2 found in both ADTKD–*SEC61A1* cell types, we assumed that the total amount of Ca²⁺ in the ER might be decreased. Accordingly, the total release of pan-organellar Ca²⁺ was induced by ionomycin and revealed an almost 20% reduction in the presence of either one of the dominant Sec61α mutations V67G or T185A (Fig 2A and B). Next, we further analyzed the continuous Ca²⁺ efflux from the ER. This efflux, often referred to as ER Ca²⁺ leakage, can specifically be unmasked by the irreversible SERCA2 inhibitor thapsigargin (Thastrup et al, 1994; Jackson et al, 1988). Intriguingly, it was the T185A mutation that caused a decreased ER Ca²⁺ leak upon treatment with thapsigargin, whereas the V67G cells behaved in an identical manner to the WT control cells (Fig 2C and D). This observation was substantiated by a series of independent measurements using the ER-targeted Ca²⁺ sensor ER-GCamP₆₋₁₅₀ (de Juan-Sanz et al, 2017). Using this protein-based Ca²⁺ sensor, we also observed a slower release of Ca²⁺ from the ER upon treatment with thapsigargin in cells carrying the Sec61α–T185A mutation (Fig S2A and B). Furthermore, the consecutive application of thapsigargin and ionomycin in a setting devoid of extracellular Ca²⁺ helped us to demonstrate that after full depletion of the ER Ca²⁺ store (achieved by thapsigargin) the remaining Ca²⁺ storage capacity of organelles

other than the ER (revealed by ionomycin) is identical for the control and mutant cells (Fig 2E and F). In other words, the reduced Ca²⁺ storage capacity of ADTKD–*SEC61A1* cells (Fig 2A and B) most likely originates in the ER (Fig 2E and F) and is probably linked to the reduced abundance of SERCA2 (Fig 1).

Other than SERCA2 we also detected a reduced level of the Orai1 protein, a Ca²⁺ channel of the plasma membrane that partakes in the so-called store-operated Ca²⁺ entry (SOCE). Upon depletion of the ER Ca²⁺ store, Orai1 is activated by STIM1 and promotes the influx of Ca²⁺ from the extracellular environment (Prakriya et al, 2006; Berna-Erro et al, 2012). To mimic the SOCE activation, depletion of Ca²⁺ from the ER was induced by thapsigargin and followed by the application of Ca²⁺ to the extracellular medium. Despite the more pronounced depletion of STIM1 in the T185A cells (Fig 1), no difference in SOCE was found for the WT and the ADTKD–*SEC61A1* cells (Fig 2G and H). Last, we wondered if the reduced Ca²⁺ leak specifically seen in T185A cells (Fig 2C–H) was a consequence of the lower amount of total Ca²⁺ (Fig 2B). Therefore, we performed a kinetic analysis and comparison of the ionomycin and thapsigargin responses in the three cell types (Fig S3). Interestingly, extending the incubation of cells in the Ca²⁺ free external solution from 1 up to 15 min caused a gradual decline of the

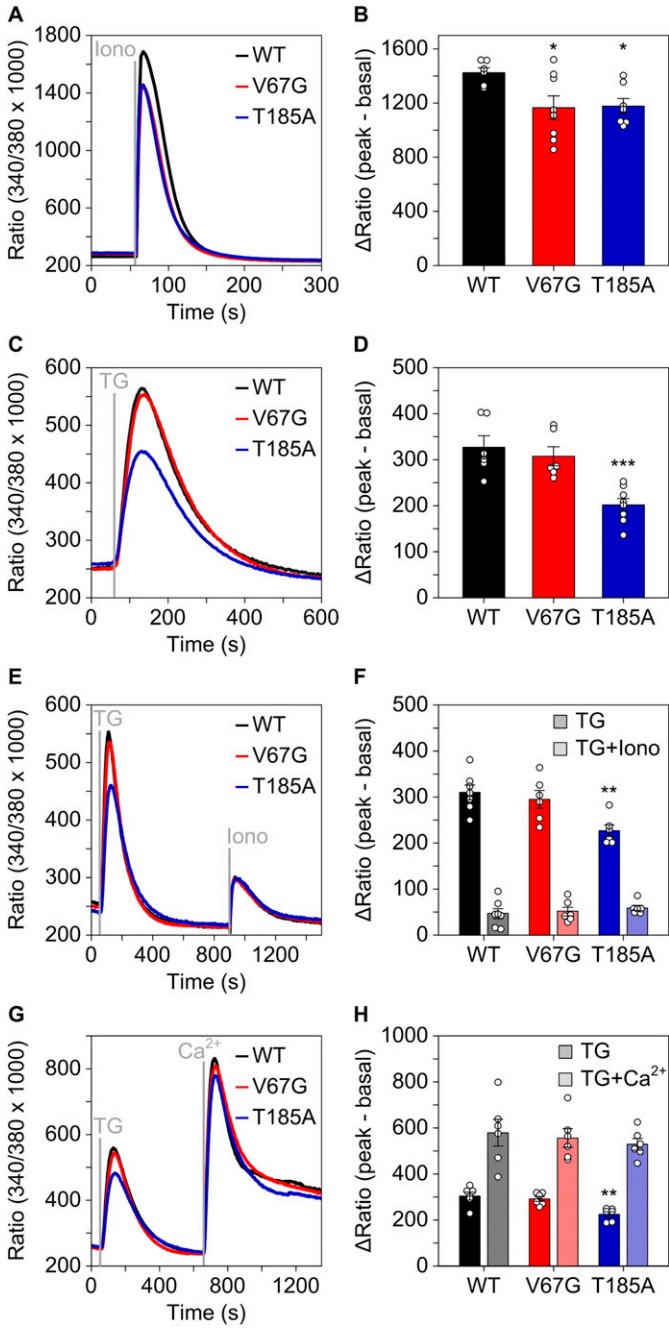

**Figure 2. Mutation-specific alterations of Ca²⁺ homeostasis in ADTKD–*SEC61A1* cells.**

**(A, B)** The WT, V67G, and T185A cell models were challenged with 5 $\mu$M of the Ca²⁺ ionophore ionomycin (Iono) in absence of extracellular Ca²⁺. Cell responses to the treatments were visualized 25 min after pre-loading cells with 3.5 $\mu$M of the ratiometric Ca²⁺ indicator Fura-2AM in medium and later quantified as ΔRatio. ΔRatio represents the difference between peak signal after treatment minus the average basal signal (30 s) before treatment. **(C, D)** Same as in (A, B), but cells were challenged with 1 $\mu$M of the SERCA inhibitor thapsigargin (TG). **(E)** The cells were treated with TG to specifically deplete the ER Ca²⁺ store, followed by the Iono treatment to visualize the sum of the remaining intracellular Ca²⁺ pools. **(F)** Individual quantification of the ΔRatio after TG treatment and after the subsequent Iono treatment. **(G)** As in (E), cells were treated with TG to deplete the ER Ca²⁺ store and to initiate the store-operated Ca²⁺ entry (SOCE). SOCE was then demonstrated by the addition of 7 mM extracellular Ca²⁺. **(H)** Individual

ionomycin response, that is, the total amount of releasable Ca²⁺ (Fig S3A–D). Yet, the amplitude of Ca²⁺ leak unmasked by thapsigargin remained unaltered in all tested cells, irrespective of the 1 or 15 min incubation in Ca²⁺ free buffer (Fig S3E–H). The kinetic data demonstrated that the reduced Ca²⁺ leak of T185A cells was independent of the lower amount of total ER Ca²⁺ and instead represented an autonomous, direct consequence of the point mutation in TMH 5.

In summary, the analysis of Ca²⁺ homeostasis of ADTKD–*SEC61A1* cells showed two distinct aberrations. First, the Sec61α mutations V67G and T185A caused a reduction of the ER Ca²⁺ content, likely related to the reduced SERCA2 abundance in both cases. And second, the T185A mutation caused a reduced Ca²⁺ leak from the ER that is unrelated to the lowered ER Ca²⁺ content and more likely a consequence of the threonine to alanine substitution in TMH 5.

## ADTKD–*SEC61A1* mutations cause a substrate-specific secretion defect of the renal marker protein renin

In addition to channeling Ca²⁺ across the mammalian ER membrane, an evolutionarily conserved function of the Sec61 complex and its ancestors is the transfer of secretory proteins across, or the insertion of membrane proteins into, a lipid bilayer (Park & Rapoport, 2012; Denks et al, 2014). Considering the diminished abundance of membrane proteins such as SERCA2 and Orai1 (Fig 1), we next addressed the issue of Sec61-mediated protein transport in more detail. To this end, we used an established in vitro procedure combining the programmed synthesis of precursor proteins in rabbit reticulocyte lysate and semi-permeabilized cells (SPCs) to reconstitute ER protein transport in a "quasi cell" (Dudek et al, 2015). Signal peptide cleavage and/or N-linked glycosylation were used to discern between non-transported precursors (p) and the transported, modified (m) forms of ³⁵S-labeled substrates. For better comparison, the transport data of each tested substrate was normalized to the WT cells, which were set to 100% and tested in parallel (Fig 3). We also expanded our set of regularly tested and well-characterized substrates by precursor proteins typically found in the kidney (Lang et al, 2012; Haβdenteufel et al, 2018). The latter included renin and uromodulin (Bleyer et al, 2007; Schweda et al, 2007; Rampoldi et al, 2011; Živná et al, 2011), which showed efficient glycosylation, that is, ER import in the presence of SPCs (Fig S4A–D).

Given that the heterozygous ADTKD–*SEC61A1* alleles are compatible with life, we did not anticipate finding a complete block of protein transport. Accordingly, we found substrate-specific impairments of Sec61-mediated protein transport, but not of the Sec61-independent membrane insertion of tail-anchored membrane proteins. Under co-translational transport conditions, that is, SPCs were present during protein synthesis, renin and preprolactin showed a reduced transport into the ER of V67G and T185A cells, whereas the transport of substrates such as invariant chain and prion protein was not affected (Figs 3A and B and S4F). Similar to the Ca²⁺ leakage (Fig 2C and D), we also observed mutation-specific

quantification of the ΔRatio after TG treatment and after the subsequent Ca²⁺ treatment. Statistical significance of differences was assessed using ANOVA and compared with the equivalent treatment in the WT control with $P < 0.05$ (*), $P < 0.01$ (**), and $P < 0.001$ (***).

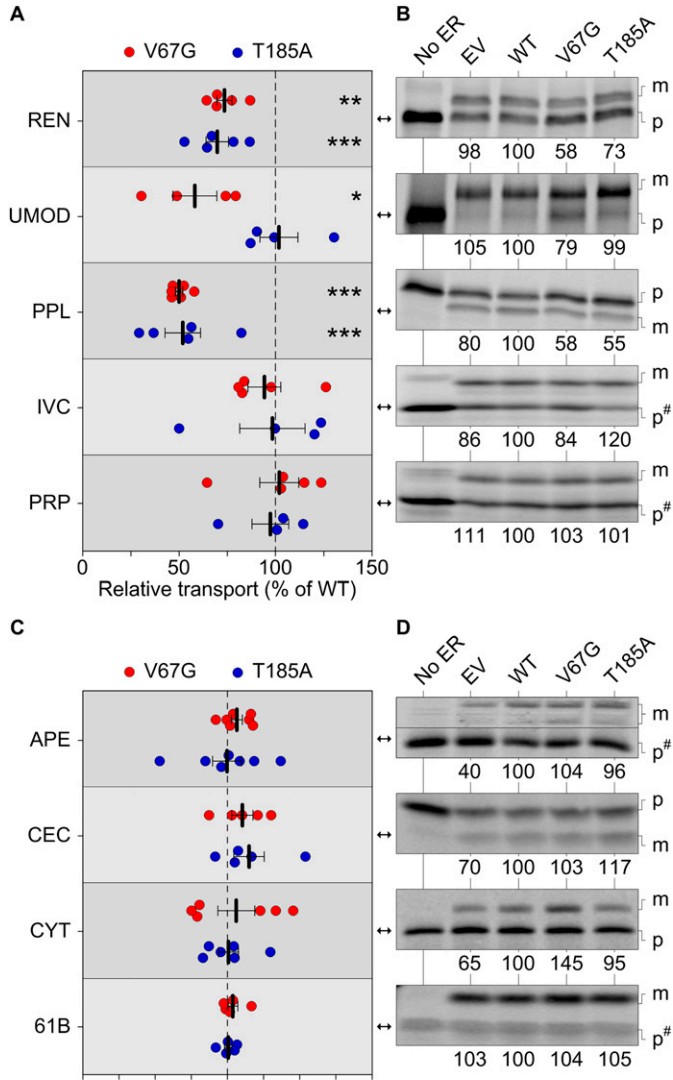

**Figure 3. Substrate-specific defects of protein transport and secretion in presence of ADTKD–SEC61A1 mutations.**
**(A, B, C, D)** Protein transport of radioactively labeled precursor polypeptides was assessed in absence and presence of functional ER. The latter refers to semi-permeabilized cells generated from the empty vector (EV) or the heterozygous wild type (WT), V67G, and T185A cell lines. Transport efficiency of each substrate was normalized to the WT cells and set to 100%. For each substrate, one of the radiograms (B, D) used for quantification of repeated measurements (A, C) is shown. **(A, B, C, D)** Substrates were tested either under co-translational conditions (A, B) with the semi-permeabilized cells being present during ribosomal protein synthesis or post-translationally (C, D), that is, the functional ER was added after completion of protein synthesis and destruction of ribosomal activity by cycloheximide and RNase A. Statistical significance of differences for the transport efficiency of each substrate was assessed using ANOVA. Cytochrome b5 (CYT), invariant chain (IVC), maturated polypeptide localized in the ER (m), precursor polypeptide (p), preproapelin (APE), preprocecropin (CEC), preprolactin (PPL), prion protein (PRP), renin (REN), Sec61β (61B), and uromodulin (UMOD). Images with the precursor polypeptide form marked as p# were compressed in one direction for easier display. The corresponding uncompressed images of IVC, PRP, APE, and 61B are shown in Fig S4F and G.

differences for protein transport. Uromodulin showed an impaired transport specifically in presence of the V67G mutations (Fig 3A and B). This finding was further supported by testing kidney disease-associated variants of renin and uromodulin (Vyleťal et al, 2006; Williams et al, 2009; Živná et al, 2009; Beck et al, 2011; Olinger et al, 2020; Živná et al, 2020). Like their wild-type counterparts, the W10R mutant of renin and the C32Y mutant of uromodulin showed impaired transport in presence of both ADTKD–SEC61A1 mutations or just the V67G mutation, respectively (Fig S4E). Other soluble (pre-proapelin and preprocecropin) and membrane-integrated (cyto-chrome b5 and Sec61β) substrates tested under post-translational transport conditions, that is, SPCs are added after protein synthesis is fully completed, were not affected by the ADTKD–SEC61A1 mutations (Figs 3C and D and S4G). Based on the variety of tested substrates one can likely exclude that the observed impairment of protein transport for renin, preprolactin, and uromodulin was related to the respective post-translational modification. Substrates showing glycosylation and/or cleavage of the signal peptide were found in both pools of affected and unaffected substrates.

Next, we asked if the maturation and/or secreted levels of renin and uromodulin would be affected in presence of the Sec61α mutations. WT and ADTKD–SEC61A1 cells were transfected with vectors encoding the wild-type uromodulin or renin for subsequent immunofluorescence and activity analyses. Evaluation of the colocalization of uromodulin with the ER luminal marker protein PDI, whose abundance was not altered in presence of ADTKD–SEC61A1 mutations (Fig 1), revealed an aberrant maturation of uromodulin only in cells with the Sec61α–V67G mutation (Fig 4A). In contrast, in WT and T185A cells, the GPI-anchored uromodulin trafficked, as expected, to the plasma membrane and showed minimal colocalization with PDI. Thus, in the presence of the Sec61α–V67G mutation, transport and trafficking of uromodulin seems to be impaired, resulting in its accumulation in the ER (Figs 3A and 4A). Regarding renin, the colocalization study also showed higher levels of overlap with the ER marker PDI, indicating a trafficking defect for renin and partial accumulation in the ER, though, in the presence of both ADTKD–SEC61A1 mutations (Fig 4B). As a consequence of the trafficking deficiency, total activity of secreted renin protein in the medium was measured by cleavage of a fluorophore-quencher pair. Proteolytic activity of the aspartyl-type endoprotease renin separates the quencher from the fluorophore and the increase in fluorescent signal of the latter serves as indicator for renin activity. As predicted, activity of renin in the medium was lower in cells carrying the V67G or T185A mutation (Fig 4C), coinciding with the reduced amount of extracellular prorenin detected by Western blot (Fig 4D).

### A signal peptide substitution rescues the hampered transport of renin in ADTKD–SEC61A1 cells

To identify the substrate-specific determinant that contributes to the impaired protein transport in ADTKD–SEC61A1 cells, we focused on the signal peptide of renin. To date, multiple dominant mutations (e.g., W10R, L16R, and C20R) in the signal peptide of renin have been found to cause ADTKD–REN, adhering to the gene-based terminology (Živná et al, 2009, 2020; Bleyer et al, 2010; Beck et al, 2011). Therefore, we replaced the signal peptide of renin by a signal peptide similar in length of the unaffected, soluble substrate preproapelin giving rise to the APE–REN construct. The signal

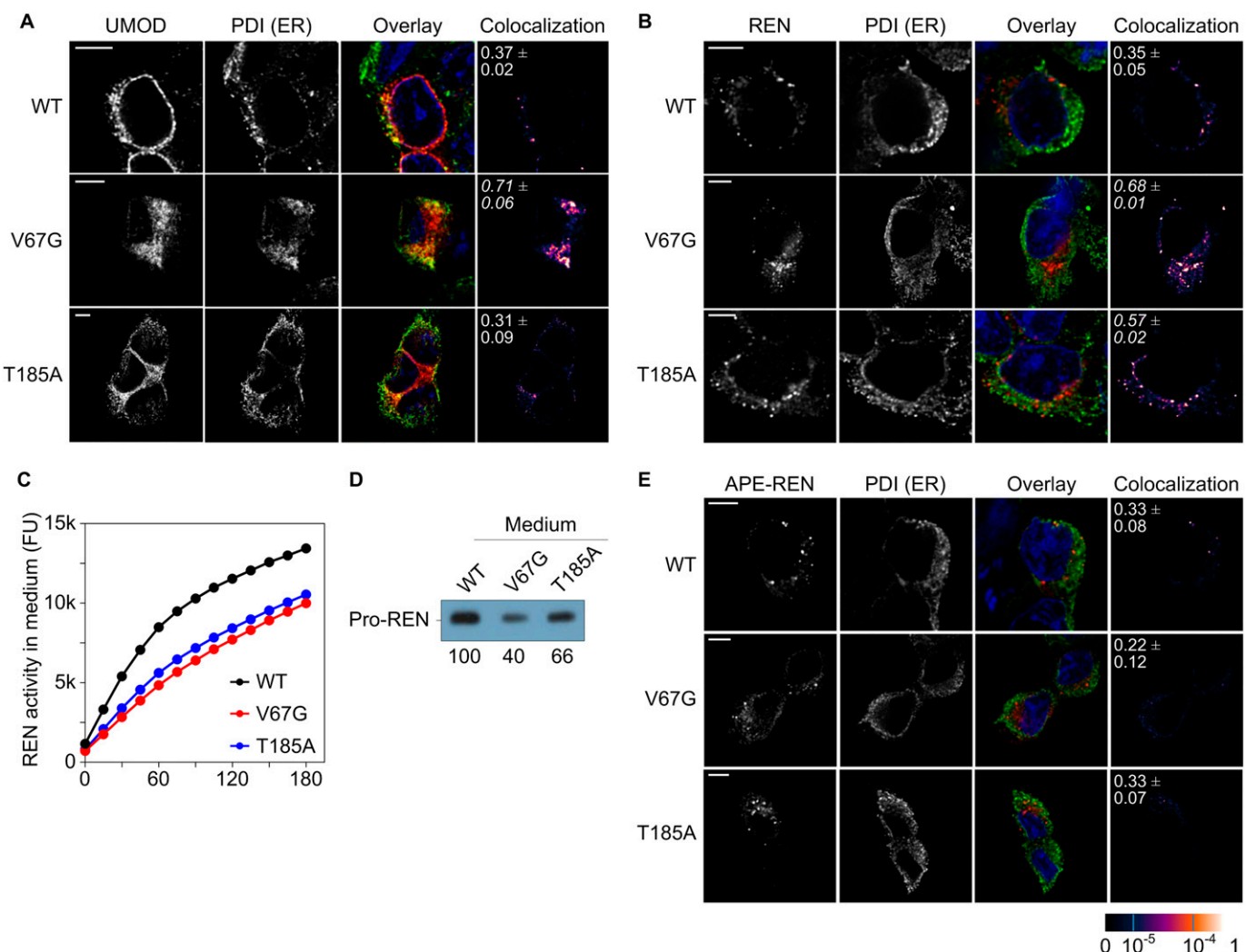

**Figure 4. Defects of uromodulin and renin maturation and secretion in presence of ADTKD–SEC61A1 mutations.**
**(A, B)** Immunofluorescence images of WT and ADTKD–SEC61A1 cells were taken 24 h after transient transfection with a uromodulin (UMOD, A) or renin (REN, B) expression plasmid. Immunostaining was used to determine the location of UMOD, REN, and an ER marker protein, protein disulfide isomerase (PDI). The degree of colocalization is shown by the overlay (UMOD/REN, red; PDI, green; nucleus, blue) and colocalization images (last column). The overlap of recorded fluorescent signals was quantified by the pseudo-colored overlap coefficient that ranges from 0 to 1 and is given as separate scale at the bottom right of the figure. A scale bar (5 µm) for each row of pictures is provided. Pearson's correlation coefficients ± SEM between red and green signals are displayed in the colocalization images as white numbers. Coefficients written in italics represent significant overlap above a 0.5 threshold. **(C)** Total activity of secreted REN was measured in medium collected from cells 24 h after transfection with a REN expressing plasmid. Cleavage of the synthetic REN substrate yielded a fluorophore and emission was recorded at 528 nm and plotted as fluorescence units (FU). Intensity of the fluorescent signal served as surrogate for total REN activity. **(D)** Western blot of medium collected from cells 24 h after transfection with a REN-expressing plasmid using a monoclonal REN antibody. **(E)** Immunofluorescence images as in (A, B) 24 h after transfection with the plasmid encoding the signal peptide exchange variant of renin, APE–REN. For APE–REN, the signal peptide of renin was exchanged by the one of preproapelin.

peptide replacement abrogated accumulation of renin's mature form in the ER (Fig 4E) and rescued its transport in the presence of either ADTKD–SEC61A1 mutation (Fig 5A and B). Conversely, substituting the signal peptide of preproapelin with the signal peptide of the affected substrate preprolactin seemed to decrease transport efficiency of the newly generated PPL–APE construct. However, the effect was statistically significant only for the V67G mutation (Fig 5A and B).

As this genetic strategy based on a signal peptide substitution could be considered a proof of principle, its effect is limited to the modified substrate of interest such as renin. An alternate chemical

approach would be to target the Sec61 complex, which might provide a broader impact on multiple substrates as well as calcium homeostasis.

## Sodium phenylbutyrate compensates for the defective renin transport and loss of ER Ca²⁺ content in ADTKD–SEC61A1 cells

To screen for a chemical modifier of the ADTKD–SEC61A1 phenotype, we decided to use cell survival as an automated readout and use the real-time cell analyzer (Fig S5A and B). We considered 10 compounds of interest (Fig S6) that cover a spectrum of biological

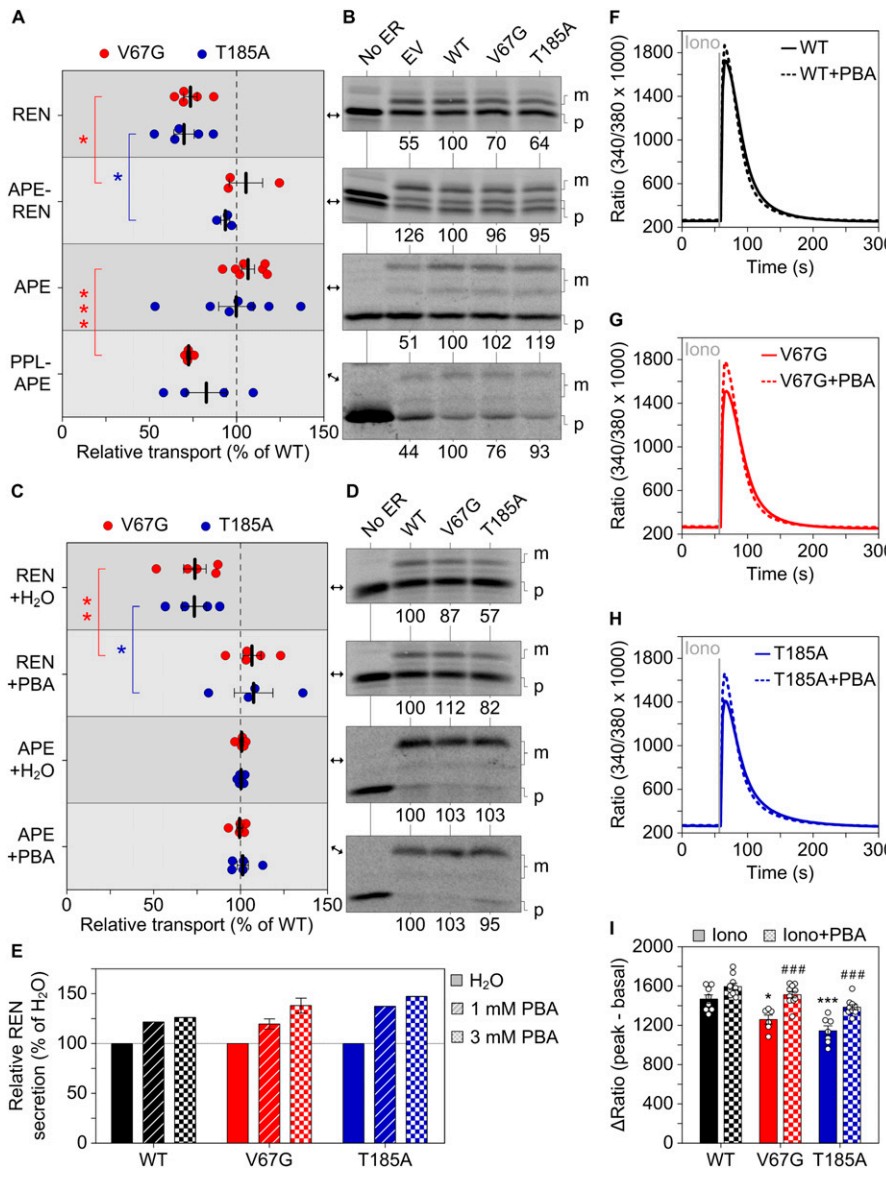

**Figure 5. Rescue of the aberrant protein transport and Ca²⁺ homeostasis.**

**(A, B)** As in Fig 3, protein transport of radioactively labeled precursor polypeptides was assessed in absence and presence of the indicated semi-permeabilized cells. The two variants APE–REN and PPL–APE were generated by replacing the innate signal sequence of REN and APE by the one from APE and PPL, respectively. REN, APE–REN, and PPL–APE were tested under co-translational conditions and APE post-translationally. For direct comparison between the variants with a signal peptide substitution (APE–REN and PPL–APE) and the original variant (REN and APE), the data gathered for the latter that were previously shown in Fig 3A and C are displayed again. However, a second radiogram is depicted here. **(C, D)** Transport efficiency of REN and APE was tested under co-translational conditions in presence of 1 mM PBA or H₂O as vehicle control. The sets of conditions connected by red or blue brackets were compared by t test. **(E)** Similar to Fig 4C, the activity of secreted REN was measured in medium collected from cells and used as readout for relative REN secretion 24 h past treatment with 1 or 3 mM PBA. 6 h past transfection, the medium was replaced by DMEM supplemented with 1 or 3 mM PBA. H₂O served as vehicle control. **(F, G, H)** As in Fig 2A, the intracellular Ca²⁺ content was determined by challenging cells with 5 µM ionomycin (Iono). 36 h before the live-cell Fura-2 measurement the cells were incubated with vehicle control or 0.5 mM PBA added to the medium. **(I)** Quantification of the ΔRatio after Iono treatment. ANOVA was used to determine the statistical significance of differences between the different vehicle control treatments ($P < 0.05$ (*), $P < 0.001$ (***)) as well as for the vehicle versus PBA treatments ($P < 0.001$ (###)), respectively.

activities such as suppression of ER stress (sodium phenylbutyrate and tauroursodeoxycholic acid), induction of ER stress (tunicamycin and dithiothreitol), activation of SERCA activity (ellagic acid and forskolin), inhibition of SERCA activity (thapsigargin) and specific inhibitors of the Sec61 complex (eeyarestatin 1), the mitochondrial ATP synthase (oligomycin A), and ribosomal protein biosynthesis (puromycin). Of the tested compounds, improved cell survival at two different concentrations was only observed with sodium phenylbutyrate (PBA), which improved cell survival of the WT and ADTKD–SEC61A1 cells at concentrations of 0.5 and 1 mM (Fig S5C–F). Although PBA is approved for the treatment of urea cycle disorders, it is also considered to possess additional pharmacological activity as a histone deacetylase inhibitor for the treatment of various types of cancer and as a chemical chaperone for the amelioration of protein misfolding diseases (Iannitti & Palmieri, 2011; Magdalena et al, 2015). We wondered if this spectrum of

activity, in particular the ability to chaperone hydrophobic domains frequently found in signal peptides and TMHs of unfolded precursor proteins, could be suited to improve the hampered protein transport in ADTKD–SEC61A1 cells. Indeed, the presence of PBA during the synthesis of renin did rescue the transport defect in ADTKD–SEC61A1 cells in comparison to the vehicle control (Fig 5C and D). The relative transport efficiency of apelin as a substrate unaffected by ADTKD–SEC61A1 mutations was not altered by PBA (Fig 5C and D). Note that in contrast to Fig 5B, apelin was tested under co-translational conditions for better comparison to the renin result, which explains the improved absolute transport efficiency of apelin (Fig 5D). Furthermore, the reduced renin secretion from cells carrying the ADTKD mutations (Fig 4C) was also improved by PBA. Using the protease activity of renin as readout for its release into the extracellular medium, a 24 h PBA treatment dose-dependently enhanced renin trafficking and secretion of the mutant and WT cells (Fig 5E).

To further explore the beneficial activity and mechanism of action of PBA in ADTKD–*SEC61A1* cells, we used sensitive $Ca^{2+}$ imaging measurements and tested the response of the mutation-specific aberrations of $Ca^{2+}$ homeostasis to PBA. Application of PBA to WT, V67G, or T185A cells did not evoke a detectable $Ca^{2+}$ transient in itself within three or 5 min (Fig S7A–F). The subsequent thapsigargin response, used to unmask the Sec61-mediated $Ca^{2+}$ leak, was slightly amplified. The magnitude of the amplification correlated with duration of the PBA pre-incubation, even in the short time window tested (Fig S7G). Both findings of the improved protein transport and secretion of renin (Fig 5C and E), as well as the amplified $Ca^{2+}$ leakage (Fig S7), indicated an immediate effect of PBA on the dynamics and gating of the Sec61 complex. We then explored whether such short-term effects could convey a benefit towards restoring protein and calcium homeostasis of ADTKD–*SEC61A1* cells over a prolonged period of time. Therefore, WT and ADTKD–*SEC61A1* cells were treated with PBA for 36 h, and the total intracellular $Ca^{2+}$ was released and measured as before using the combination of ionomycin and Fura-2 (Fig 5F–I). The corresponding quantification in Fig 5I shows that PBA elevated the $Ca^{2+}$ pool in both ADTKD–*SEC61A1* cell types and restored the cellular $Ca^{2+}$ content to that of the WT control cells. Long-term PBA treatment also increased the $Ca^{2+}$ level of WT (+9%) cells, but not as much as in the V67G (+20%) or T185A (+21%) cells.

### Structure–function relationship of the ADTKD–*SEC61A1* mutations V67G and T185A

One of the central features of the Sec61 complex is the productive gating of its polypeptide- and calcium-permeable pore, that is, the transition between the closed and the open state. High-resolution structures are available for both conformations of the Sec61 complex and provide insights into the positioning of the V67 and T185 residue (Voorhees et al, 2014; Voorhees & Hegde, 2016). Both residues are located in different domains of the Sec61 complex, and the corresponding mutations V67G and T185A likely interfere differently with the Sec61-mediated protein transport and calcium leakage (Fig 6A and B). Amino acid V67 sits at the tip of the plug domain that occupies the luminal funnel of the Sec61 complex in the closed conformation. In this position, V67 seems to support the hydrophobic residues of the so-called pore ring (I82, V85, I179, I183, I292, and L449) to seal the channel in its closed conformation (Fig 6A). During opening of the Sec61 complex by a signal peptide and its "transitory integration" into the lateral gate, the signal peptide might come in contact with the V67 residue or the plug domain which is repositioned in the process of opening. Amino acid T185 sits in the middle of TMH 5 of the Sec61α protein and like V67 on the luminal side of the constriction zone formed by the pore ring. In the closed state, the side chain of T185 faces away from the central constriction (Voorhees et al, 2014; Bolar et al, 2016). However, upon opening of the Sec61 complex during protein translocation TMH 5 rotates and T185 faces more towards the central, open pore (Fig 6B). In this configuration T185 lies opposite to the lateral gate and is part of the tunnel wall that shapes the pore. As such, T185 might contact the polypeptide in transit.

Gating of the Sec61 complex is strongly influenced by incoming signal peptides that act as allosteric regulators. Therefore, we compared the N-terminal signal peptide or its targeting equivalent, the first TMH, of substrates affected and unaffected in the presence of the V67G or T185A mutation (Fig 6C). We considered proteins from the transport assay (Fig 3) and the Western blot analysis (Fig 1) as potential candidates. Looking at the primary structure and $\Delta G_{app}$ values of the targeting signals, two features emerged that might separate the small pool of affected and unaffected substrates. First, an early negatively charged amino acid in the N-region of the signal peptide (REN, PPL, and STIM1) or the cytosol-facing end of the first TMH (SERCA2) appeared more frequently in the affected versus the unaffected substrates (Fig 6C). Second, except for uromodulin, the $\Delta G_{app}$ value of affected targeting signals was above the average $\Delta G_{app}$ of human signal peptides (+1.74; Fig 6D) and first TMHs (−0.51; Fig 6E). The above-average $\Delta G_{app}$ value of affected candidates indicates their lower propensity for efficient membrane integration. However, several points should be noted. One, the tested preprolactin was from *Bos Taurus* and differs from its human counterpart. The signal peptide of human preprolactin lacks any negative charge and has a lower $\Delta G_{app}$ value (+3.36). Thus, transport and secretion of human preprolactin might not be affected in ADTKD–*SEC61A1* patients. Two, the signal peptide mutation of renin, REN-W10R, showed an overall reduced transport fidelity and impaired transport into the ER of V67G and T185A cells (Fig S4E). The REN-W10R signal peptide mutation preserves the early negative charge and has an even higher $\Delta G_{app}$ value (+3.52) compared with the REN signal peptide. When the negatively charged aspartate in position 2 was replaced by an arginine the resulting REN-D2R variant with a $\Delta G_{app}$ value of +1.63 did not show a transport defect in vitro (Fig S4E) or accumulation in the ER using immunofluorescence (Fig S8). Three, we included STIM1 and Orai1 in this compilation that, similar to SERCA2, showed a reduced protein abundance in one or both ADTKD–*SEC61A1* cell types (Fig 1).

In summary, our characterization of a cellular model for ADTKD–*SEC61A1* highlighted distinct changes of protein and $Ca^{2+}$ homeostasis in the presence of the dominant V67G and T185A mutation of the Sec61α protein. These findings align well with the cellular placement and function of Sec61α as central component of the Sec61 complex, and hence, the protein translocase of the human ER known to be a conduit for both polypeptides and ions. The specific impairments found in protein transport (e.g., renin) and protein abundance (e.g., SERCA2) could be the direct cause for the reduced $Ca^{2+}$ content of the ER. This conclusion was further supported by the identification of PBA as small molecule that can reverse the impairment of renin secretion and $Ca^{2+}$ handling in the presence of ADTKD–*SEC61A1* mutations. It is also important to note that we identified mutation-specific variations such as the reduced $Ca^{2+}$ leakage in presence of the T185A mutation, which might help to explain differences in the clinical and laboratory features of ADTKD–*SEC61A1* affected patients.

## Discussion

In this study, we characterized the cellular malfunctions that occur in presence of the Sec61α mutations V67G and T185A causing the monogenetic kidney disease ADTKD. Our analysis demonstrated a

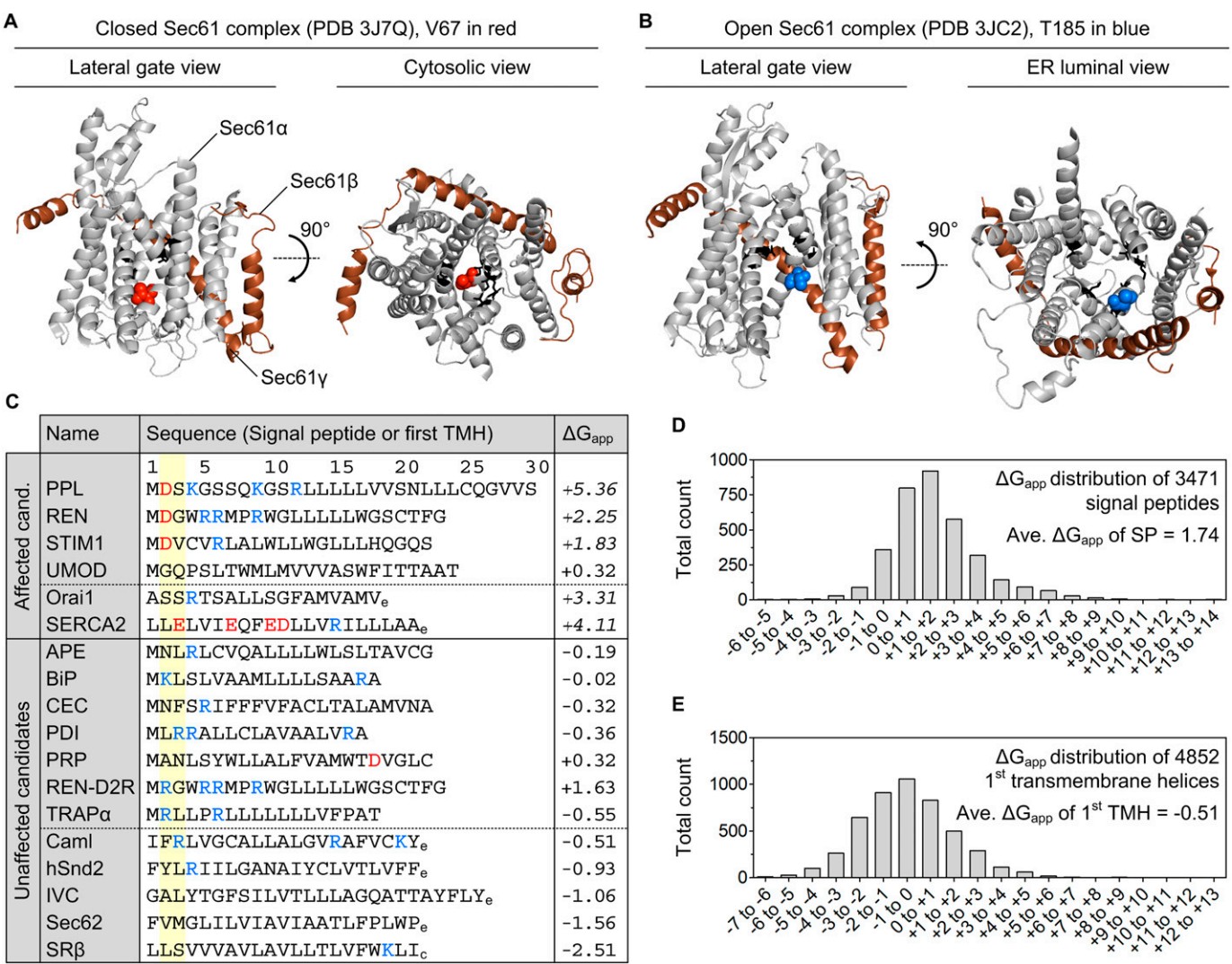

**Figure 6. Biophysical attributes potentially differentiating substrates affected or unaffected in presence of ADTKD–*SEC61A1* mutations V67G and T185A.**
**(A, B)** Structural display of the closed (PDB 3J7Q) and open (PDB 3JC2) conformation of the Sec61 complex (Voorhees et al, 2014; Voorhees & Hegde, 2016). Sec61α is shown in light grey, Sec61β (for comparison always shown on the right side) and Sec61γ are shown in brown. Pore ring residues (I82, V85, I179, I183, I292, and L449) are presented in black as stick model. Residues V67 (A) and T185 (B) are highlighted as spheres in red and blue, respectively. **(C)** Listed are the two categories of substrates that were identified in the transport assay (Fig 3) or Western blot analysis (Fig 1) as affected or unaffected candidates in presence of the V67G or T185A mutation of the Sec61α protein. The sequence provided for each candidate is the signal peptide (SP) or, if absent in a protein, its targeting equivalent, the first transmembrane helix (TMH). Hence, each category (affected and unaffected) is split by a dotted line. Above this line are candidates carrying an N-terminal SP and below are membrane proteins without a SP. Positively (K, R) and negatively (D, E) charged amino acids are indicated in blue and red, respectively, as well as the ΔG$_{app}$ value for the given sequence (Hessa et al, 2005, 2007). **(D, E)** ΔG$_{app}$ values above the average for human signal peptides (D) or the first TMH of membrane proteins (E) are indicated in italic. **(D, E)** Distribution of the ΔG$_{app}$ values for 3,471 human signal peptides (D) and the first TMH of 4,852 bi- or polytopic membrane proteins (E). Sequences for analysis were extracted from the UniProt database (release version 2019_10 and later) and histograms were generated including the average ΔG$_{app}$ value for SPs (+1.74) and first TMHs (−0.51). Of note, SPs and TMHs annotated with a length shorter than 8 aa or longer than 40 aa were excluded from the ΔG$_{app}$ analyses based on the restrictions of the algorithm (https://dgpred.cbr.su.se/index.php?p=home). Subscripted characters at the end of a TMH sequence indicate the ER luminal/extracellular (e) or cytosolic (c) orientation of this terminus.

substrate-specific impairment of protein transport with a small set of precursors being affected in presence of the ADTKD–*SEC61A1* mutations. In addition to the heterologous substrate PPL from *Bos taurus*, the two renal marker proteins of human descent, renin (REN), and uromodulin (UMOD), showed a partially reduced transport and secretion in the presence of one or both mutants (Figs 3 and 4). Previous assessments of REN biosynthesis identified mutations in the REN signal peptide such as L16R or ΔL16 that cause

a reduced ER translocation without differences in the amount, shape, and cellular localization of REN-containing granules in the cytosol (Živná et al, 2009). These data highlight the importance of a functional REN signal peptide for efficient translocation and corroborate, however, indirectly, our observation of rescuing the impaired REN transport in ADTKD–*SEC61A1* cells by a signal peptide substitution or the D2R charge reversal (Figs 5A, S4E, and S8). In case of UMOD and REN the immunofluorescence imaging revealed that

the proteins accumulate in the ER or being retained at the mutant Sec61 complex of one or both types of ADTKD–*SEC61A1* cells (Fig 4). The ER luminal retention could also represent a secondary phenotype resulting from the impaired biogenesis of an unspecified, but Sec61-dependent factor that is required for further maturation. For example, UMOD, which was only retained in the ER of the V67G cells, is a prominent GPI-anchored protein synthesized predominantly in tubular epithelial cells of the thick ascending limb in nephrons (Rampoldi et al, 2011). Future endeavors could aim at monitoring the maturation of other GPI-anchored proteins by microscopy as well as protein abundance levels of the ER localized GPI transamidase subunits in ADTKD–*SEC61A1* cells, particularly V67G. Alternatively, the observed accumulation of REN and UMOD in the ER could be related to a missing prohormone convertase required for further processing.

Strikingly, the partial transport and maturation defect of REN and UMOD in the ADTKD–*SEC61A1* setting resembles the disease pattern observed for numerous mutations identified in the cleavable signal peptide or mature domains of REN and UMOD. Six mutations in the *REN* gene and more than 100 mutations in the *UMOD* gene have been identified that cause a progressive decline in kidney functionality and are diagnosed as ADTKD–*REN* and ADTKD–*UMOD*, respectively (Devuyst et al, 2019; Schaeffer et al, 2019; Olinger et al, 2020). Although mutations in the REN signal peptide correlate with a reduced REN translocation (Živná et al, 2009), many of the UMOD mutations cause accumulation of the protein in the ER and, later on, induce an ER stress response (Williams et al, 2009; Vyletal et al, 2010). Both of those clinical features, reduced REN secretion and ER retention of UMOD, were recapitulated in the ADTKD–*SEC61A1* cells upon transient transfection of *REN* and *UMOD* expressing plasmids (Fig 4). Thus, the manifestation of a kidney disease pattern in patients carrying the Sec61α–V67G or Sec61α–T185A mutation could very well be explained by the substrate-specific retention of REN and eventually UMOD in or at the ER, thereby hampering their efficient secretion. Given that (i) the expression of *REN* and *UMOD* is almost exclusively restricted to cells of the nephron and (ii) these proteins are involved in renal function, mass, or development we propose that the known cases of ADTKD–*SEC61A1* phenocopy to a large extent the clinical hallmarks of ADTKD–*REN* and, in case of V67G, ADTKD–*UMOD* (Gribouval et al, 2005; Castrop et al, 2010; Gomez & Sequeira-Lopez, 2016; Steubl et al, 2016; Pivin et al, 2018). In addition, impaired nephrogenesis was also recapitulated in zebrafish models expressing the V67G and T185A variants of *SEC61A1* or the L381P substitution of *REN*, what further strengthens the idea of a phenocopy (Bolar et al, 2016; Schaeffer et al, 2019).

Our data also shed light on another aberrance associated with the ADTKD–*SEC61A1* mutations, dysregulation of $Ca^{2+}$ homeostasis (Fig 2). Both mutations, V67G and T185A, caused reduced intracellular $Ca^{2+}$ content that we attributed to a reduced $Ca^{2+}$ storage in the ER. One of the main driving factors controlling $Ca^{2+}$ storage in the ER is SERCA2, the most prominent P-type ATPase of the ER membrane responsible for $Ca^{2+}$ refilling. Corresponding to the reduced ER $Ca^{2+}$ content shown by Fura-2 measurement, Western blot analysis identified SERCA2 as one of the few proteins with reduced overall abundance (Fig 1). Reduced $Ca^{2+}$ content of the ER in V67G and T185A cells could also imply an impaired $Ca^{2+}$ storage capacity. Many chaperones of the ER, including BiP and PDI, are

characterized as high capacity $Ca^{2+}$ buffering proteins and as such shape the $Ca^{2+}$ storage capacity of the organelle (Coe & Michalak, 2009). As the steady-state protein levels for BiP and PDI remained unchanged (Fig 1), we attribute the lowered ER $Ca^{2+}$ content to a diminished SERCA2 abundance.

Much to our surprise, analysis of the $Ca^{2+}$ leak from the ER showed, despite reduced SERCA2 levels in both ADTKD–*SEC61A1* cell types, a reduced efflux of $Ca^{2+}$ specifically in the T185A background (Fig 2C–H). As we did not find additional regulators of ER $Ca^{2+}$ homeostasis being affected, the dampened $Ca^{2+}$ efflux in T185A cells is likely a direct effect of the threonine to alanine substitution within the Sec61 complex. Upon opening of the Sec61 complex TMH 5 rotates and T185 faces more towards the central, open pore (Fig 6A and B). In this open conformation, the hydroxyl group of threonine could contribute to the ion conductance of the Sec61 complex and channel $Ca^{2+}$ towards the open pore by providing a binding site for $Ca^{2+}$ or a substitute for the $Ca^{2+}$ water shell. Interestingly, it is upon termination of protein translocation that the Sec61 complex transitions from the open to the closed state and becomes temporarily permeable to $Ca^{2+}$ (Wirth et al, 2003). In addition to losing a transient $Ca^{2+}$ binding spot, the T185A mutation could also affect the gating characteristics of the channel. This could provide an explanation for its defects in protein and $Ca^{2+}$ handling. With regard to the V67G mutation, the residue V67 is flanked by the positively charged R66. In this context, the smaller side chain and helix breaking potential of glycine might increase the accessibility of a signal peptide to the R66 residue. Thereby, the negative charge in the signal peptides of REN and PPL could cause an unproductive interaction (D2–R66) that hampers efficient opening of the Sec61 complex carrying the V67G mutation.

Last, multiple defects observed in ADTKD–*SEC61A1* cells could be reversed by the small molecule PBA. Whereas the short-term treatment with PBA rescued the impaired REN transport in vitro (Fig 5C), longer treatments of 20 or 36 h were able to restore REN secretion from live cells (Fig 5E) or normalize the $Ca^{2+}$ concentration of the ER (Fig 5I), respectively. PBA is an FDA approved drug for the treatment of infants as well as adults suffering from urea cycle disorders that can cause life-threatening hyperammonemia. The drug is usually taken orally with a defined daily dose of 20 g/day by adults. PBA also appears to be a multi-purpose agent showing additional pharmacological activity as histone deacetylase inhibitor (HDACi) as well as a chemical chaperone capable of influencing gene expression and proteostasis, respectively (Iannitti & Palmieri, 2011). Interestingly, expression of the human *REN* gene relies on efficient acetylation of a cAMP-response element found in the proximal promotor and a distant enhancer and thus could be increased by an HDACi like PBA. As a molecular chaperone PBA might have the ability to interact with and stabilize hydrophobic domains frequently found in signal peptides and TMHs. PBA could support such targeting peptides and provide a longer dwell time for their interaction with the translocon. Alternatively, PBA might act as catalyst that shifts the free energy barrier of the Sec61 complex to improve the transport of substrates with infrequent features, such as the high $\Delta G_{app}$ value and an early negative charge in the signal peptide or first TMH. Regardless of the mechanism of action, PBA reverses some of the aberrations that could be pathognomonic for ADTKD–*SEC61A1*. With the earlier proposition of a phenocopy

between ADTKD–*SEC61A1* and ADTKD–*REN* in mind, PBA might also be an interesting drug to consider for cases of ADTKD–*REN*. However, we would also like to point out the ineffectiveness of PBA as a treatment in two ADTKD–*UMOD* mouse models. In both models, PBA showed predominantly HDACi activity that increased the expression of the mutant and wild type *UMOD* gene rather than chaperone activity (Kemter et al, 2014). Importantly, like ADTKD–*REN*, ADTKD–*SEC61A1* is also a disease with a dominant pattern of inheritance. Thus, after genetic testing, PBA could be a potential early therapy for children with ADTKD–*SEC61A1* before severe renal insufficiency occurs, which is considered a contraindication for the use of PBA.

Taken together, we provide insights into the structure–function relationship of the Sec61 complex and a potential explanation for the kidney-specific manifestation in the case of the ADTKD–*SEC61A1* causing mutations V67G and T185A. It appears that the aberrations originate in a substrate-specific impairment of protein transport for REN and SERCA2 that further propel alterations of $Ca^{2+}$ homeostasis.

# Materials and Methods

## Materials

The CP-BU Medical X-ray blue film, SensoLyte 520 renin assay kit, Clarity Western ECL substrate, GelRed nucleic acid dye, and digitonin were purchased from Agfa, AnaSpec, Bio-Rad, Biotium, and Calbiochem, respectively. EGTA, glucose, Hepes, and powdered milk were obtained from Carl Roth, the DMEM/F12 medium from Cytiva, the Fomadent solution set from Foma Bohemia, and geneticin G418, PBS, and Trypsin–EDTA from Gibco. 3.5 and 6 cm dishes as well as T12 flasks were ordered from Greiner Bio-One, ionomycin and DAPI from Invitrogen, the MycoAlert Detection Kit plus the MycoAlert Assay Control Set were obtained from Lonza Cologne, and Merck products included Amicon Ultra 0.5 ml Centrifugal Filters 10K, eeyarestatin 1, glycerol, magnesium acetate, magnesium chloride, phenylbutyric acid, polyvinylidene fluoride (PVDF) membrane, potassium chloride, sodium chloride, and Tris. 25-mm cover slips were purchased from Neolab, streptomycin sulfate from PAA, and [$^{35}$S]-methionine from Perkin Elmer. Promega delivered alkaline phosphatase, FuGENE HD, PFU polymerase, and reticulocyte lysate both the nuclease-treated as well as TNT-coupled version. The DNeasy Blood & Tissue Kit and the QIAquick Gel Extraction Kit were purchased from QIAGEN. Cycloheximide, DNase I, NP40, PMSF, and RNase A were obtained from Roche, SDS and bromophenol blue from Serva, antipain, chymostatin, DMSO, FBS, leupeptin, 2-mercaptoethanol, penicillin G, pepstatin A, protease inhibitor cocktail, and puromycin from Sigma-Aldrich. Other reagents (BamHI, buffer G, Fura-2AM, Lipofectamine TM2000, PageRuler prestained protein ladder, potassium acetate, Sac II, T4 DNA ligase, and thapsigargin) were bought from Thermo Fisher Scientific.

## Generation of HEK293 cells stably expressing C-terminal FLAG–tagged *SEC61A1*

$3 \times 10^5$ HEK293 cells were seeded into six-well plate and maintained in DMEM high-glucose medium supplemented with 10% (vol/vol) FBS, 100 U/ml penicillin G, and 100 mg/ml streptomycin sulfate. Within 24 h, cells were transfected with Lipofectamine 2000 (Invitrogen) with 4 µg of DNA. 3 d after transfection, cells were trypsinized, diluted, and cultured in selective medium containing 0.8 mg/ml G418. *SEC61A1*-FLAG-expressing clones were selected with PCR, sequencing, and Western blot analyses.

## Cultivation, proliferation, and genetic manipulation of mammalian cells

Stably transfected HEK293 cells were cultured in DMEM/F12 medium containing 10% FBS and 0.8 mg/ml geneticin G418 in a humidified environment with 5% $CO_2$ at 37°C. Cells were maintained in T25 flasks and passaged thrice a week. For passaging, cells were carefully rinsed with PBS (pH 7.2), detached using 0.05% Trypsin–EDTA, and a fourth of the cell suspension was further cultivated. Cell numbers were routinely determined using the Countess Automated Cell Counter (Invitrogen) according to the manufacturer's instructions. In addition, cell proliferation was monitored in real time and noninvasively using the RTCA biosensor technology in the xCELLigence SP system (Roche) using 96-well E-plates (Omni Life Science). After thawing, each batch of cells was tested for potential mycoplasma contamination using the MycoAlert Detection Kit and the MycoAlert Assay Control.

## Genomic DNA purification and Sanger sequencing of transgenic *SEC61A1* variants

Approximately $4 \times 10^6$ cells were harvested and processed using the DNeasy Blood & Tissue Kit. During cell lysis RNA was removed using 170 µg/ml RNase for 2 min at room temperature. After the column-based purification, DNA was eluted in 200 µl water. During the procedure cellular DNA is usually fragmented to pieces of 30–50 kb in size. To determine quantity and quality 1 µl of DNA was used for UV–Vis spectrophotometry (Nanodop-1000, Peqlab) and analyzed on a 0.5% agarose gel. For visualization, agarose gels were stained for 4 h with GelRed nucleic acid dye diluted in water. Transgenic *SEC61A1* loci were amplified using a PFU polymerase driven PCR and a specific primer pair (#1_fwd: TAATACGACTCACTATAGG; FLAG_rev: CTTGTCGTCATCGTCTTTGTAGTC) for 35 cycles. The amplified 1.5 kb PCR product was analyzed on a 1% agarose gel and send for Sanger sequencing (LGC Genomics) using the primers mentioned before and two additional primers (#2_fwd: GAAACCATCGTATGGAAGG; #3_rev: CTCGGAAGCCCTGGAAATAG).

## Generation of the APE–REN signal peptide swap construct

Using a fwd (CTTGACCGCGGTGTGTGGACTCCCGACAGACACCACCAC) and rev primer (GCGCGGATCCCTAGCGGGCCAAGGCGAAG) prorenin was amplified from the pCR3.1 vector encoding human renin via PCR with overhangs encoding a SacII and BamHI restriction site in the 5' and 3' end, respectively. Taking advantage of a SacII restrictions site within the apelin signal peptide, the pTNT vector encoding human apelin as well as the prorenin PCR product were digested with SacII/BamHI in buffer G for 2 h at 37°C, separated on a 1% agarose gel, stained with GelRed nucleic acid dye for 4 h and bands of vector (2,617 bp) as well as PCR product (1,164 bp) were extracted, purified,

and eluted in 10 $\mu$l EB buffer using the QIAquick Gel Extraction Kit. Before the gel run, the double digested vector was treated with alkaline phosphatase for 1 h at 37°C. Cleaned up PCR product and vector were mixed in a 3:1 (vol/vol) ratio and ligated using 0.2 U/$\mu$l T4 DNA ligase for 90 min at room temperature.

## Western blot analysis of cell lysates

Cell pellets were harvested and 25,000 cells/$\mu$l were solubilized in lysis buffer containing 10 mM Tris/HCl, pH 8.0, 10 mM NaCl, 3 mM $MgCl_2$, 0.5% (vol/vol) NP40, 0.1 mM PMSF, 0.1 mg/ml DNAse, and 0.1% (vol/vol) proteinase inhibitor cocktail PLAC. PLAC is a mixture of pepstatin A, leupeptin, antipain, and chymostatin (each 3 mg/ml) dissolved in DMSO. Resuspended cell pellets were incubated at 37°C for 30 min to digest DNA and subsequently diluted to 20,000 cells/$\mu$l with 5x Laemmli sample buffer (300 mM Tris/HCl, pH 6.8, 50% [vol/vol] glycerol, 10% [wt/vol] SDS, 25% [vol/vol] 2-mercaptoethanol, and 0.05% [wt/vol] bromophenol blue). Usually, 10–15 $\mu$l of sample solution were separated via SDS–PAGE before blotting on a PVDF membrane at 400 mA for 3 h. After blocking with 3% (wt/vol) powdered milk, membranes were incubated with specific mono- or polyclonal primary antibodies. Apart from PDI and Grp170 where the purified canine protein was used for immunization, polyclonal rabbit antibodies were also raised against short peptides of human Sec61$\alpha$ (KEQSEVGSMGALLF), Sec61$\beta$ (PGPTPSGTN), SR$\beta$ (MASADSRRVADGG), Caml (SQRRAELRRRKLLMNSEQRINRIMGF, and HRPGSGAEEESQTKSKQQDSDKLNSL), hSnd2 (KRQRRQERRQMKRL), BiP (EEEDKKEDVGTV), Sec62 (MAERRRHKKRIQ), and TRAP$\alpha$ (LPRKRAQKRSVGSDE) as well as the FLAG antigen (DYKDDDDK). The peptides carried an additional N- or C-terminal cysteine for cross-linking to KLH. Rabbit polyclonal antibodies against Rpn1 and Stim1 (Proteintech) were kindly provided by Christopher Nicchitta (Duke University Medical Center) and Barbara Niemeyer (Saarland University), respectively. The rabbit polyclonal antibody against Orai1 and the murine monoclonal antibodies against SERCA2 and $\beta$-actin were purchased from Sigma-Aldrich. Primary antibodies were visualized using ECL Plex goat anti-rabbit IgG-Cy5 (VWR) or ECL Plex goat anti-mouse IgG-Cy3 (GE Healthcare) and the Typhoon-Trio imaging system in combination with the ImageQuant TL software version 7.0 (GE Healthcare).

## Western blot analysis of extracellular medium

8 × 10$^5$ HEK293 cells stably expressing the WT, V67G, or T185A variant of *SEC61A1*-FLAG were cultured in FBS and phenol red free medium. After 24 h cells were transiently transfected with 4 $\mu$g of an expression plasmid encoding for wild type renin using 10 $\mu$l of Lipofectamine TM2000. 24 h after lipofection, the medium was collected and mixed with protease inhibitor cocktail in a ratio of 100:1 (vol/vol). The medium was centrifuged first at 800$g$ for 5 min and then at 15,000$g$ for 5 min to remove residual cells and cellular debris. Supernatants were concentrated on Amicon Ultra - 0.5 ml Centrifugal Filters 10K to minimal volume (about 20 $\mu$l) according to the manufacturer's instructions. Volumes of concentrated media equivalent to 250,000 cells were dissolved in 6× SDS sample buffer (350 mM Tris Base, 10% [wt/vol] SDS, 6% [vol/vol] 2-mercaptoethanol, 30% [vol/vol] glycerol, and 0.012% [wt/vol] Bromophenol Blue, pH 6.8),

denatured at 100°C for 5 min and resolved on a 4% stacking and 10% separating gel at 75 and 150 V, respectively, in MightySmall II SE260 or SE640 apparatus (Hoefer). Proteins were then transferred to a methanol-activated PVDF membrane using the semi-dry blotting apparatus PHERO-Multiblot (Biotec-Fischer) at 0.6 mA/cm$^2$ for 60 min. Membranes were blocked in phosphate-buffered saline with 0.1% Tween-20 (PBS-T) and 5% BSA and probed with rabbit pre-prorenin (288-317) (Yanaihara Institute Inc.) diluted 1:3,000 in PBS-T with 0.1% BSA followed by HRP-conjugated Goat anti-Rabbit IgG (H+L) secondary antibody (Thermo Fisher Scientific) at 1:10,000 dilution. Membranes were incubated with Clarity Western ECL Substrate according to manufacturer's instructions. Signal was captured on CP-BU Medical X-ray blue film and developed by Fomadent solution set.

## Live-cell imaging of cytosolic Ca$^{2+}$ levels

Fluctuations of cytosolic Ca$^{2+}$ transients were measured using the ratiometric Ca$^{2+}$ indicator Fura-2. HEK cells were freshly passaged 1:2 into a T25 flask. 24 h later, cells were harvested, counted, and seeded in a 6-cm dish with ~5 × 10$^5$ cells/4 ml and incubated for 48 h in a humidified environment with 5% $CO_2$ at 37°C. Each 6 cm dish contained two 25 mm cover slips (thickness 0.24 mm) that were transferred separately into 3.5-cm dishes filled with 1 ml pre-warmed medium. Cells were loaded with 3.5 $\mu$M Fura-2AM for 25 min in the dark at room temperature, mounted into the specimen holder and washed twice with calcium-free buffer containing 140 mM NaCl, 5 mM KCl, 1 mM $MgCl_2$, 0.5 mM EGTA, and 10 mM glucose in 10 mM Hepes/KOH, pH 7.35. If not stated otherwise, cells were treated with 1 $\mu$M thapsigargin, 5 $\mu$M ionomycin, 7 mM Ca$^{2+}$, or 0.5 mM PBA during the ratiometric measurements that were carried out in 3 s intervals. During each interval, data were collected using the iMIC microscope and the polychromator V (Till Photonics) by alternate excitation at 340 and 380 nm and recording the emitted fluorescence at 510 nm. Recorded Fura-2 signals represented the F340/F380 ratio, whereas F340 and F380 correspond to the background-subtracted fluorescence intensities at the 340 and 380 nm wavelength, respectively. The iMIC was equipped with a Fluar M27 lens with a 20× magnification and a 0.75 numerical aperture (Carl Zeiss) plus the iXon+camera (Andor Technology). Per run, 50 cells were evaluated and analyzed using Excel of the Microsoft 365 package. The average of 50 cells was considered as one data point.

## Live-cell imaging of ER luminal Ca$^{2+}$ dynamics

The genetically encoded, low affinity Ca$^{2+}$ sensor GCaMP$_{6-150}$ was used to measure the ER Ca$^{2+}$ dynamics in response to 1 $\mu$M thapsigargin (de Juan-Sanz et al, 2017). HEK cells were seeded on a 25-mm cover slip (thickness 0.17 mm) in a 3.5 cm dish using 3 × 10$^5$ cells/2 ml. After 24 h, the cells were transiently transfected with 2 $\mu$g of the GCaMP$_{6-150}$ encoding plasmid using 5 $\mu$l FuGENE HD and incubated for another 24 h in absence of any antibiotics. Cells were mounted into the specimen holder and washed twice with calcium-free buffer. Measurements were performed in 3 s intervals using 2 × 2 binning and 50 ms exposure times, and the GFP filter setting (excitation 480 nm, emission 510 nm) using the iMIC microscope in combination with the polychromator V (Till Photonics), the Fluar M27 lens (Carl Zeiss) with a 100× magnification (oil immersion) and a

0.75 numerical aperture plus the iXon+camera (Andor Technology). After 1 min, thapsigargin was added and the ER calcium depletion was monitored. The data were used to calculate the decay constant and the time till 50% of the ER calcium content is released ($T_{0.5}$). Data were plotted as normalized signal intensities with $I = (F-F_b)/(F_0-F_b) = F'/F'_0$. F represents the measured fluorescence signal at a given time, $F_b$ the average of the remaining basal fluorescence after ER calcium depletion by thapsigargin (average of the last 20 frames measured), and $F_0$ the average fluorescence intensities before application of thapsigargin (average of the frames 5–15).

### Preparation of SPCs

SPCs were used as a functional, cellular ER fraction. The preparation of SPC was performed similar to previously published procedures with two minor adjustments accounting for the smaller size of HEK cells compared with HeLa (Lang et al, 2012; Dudek et al, 2013). One, for permeabilization, 0.5 $\mu$l digitonin (40 mg/ml in DMSO) per $10^6$ cells was used. Two, during the final re-isolation, the SPC pellet was adjusted to 50,000 cell equivalents/$\mu$l in SPC buffer (110 mM potassium acetate, 2 mM magnesium acetate, 20 mM Hepes/KOH, pH 7.2). Later on, equivalent cell numbers of simultaneous SPC preparations were confirmed by SDS–PAGE and Coomassie protein staining.

### In vitro protein transport using SPC

Protein transport was reconstituted and performed with slight modifications to earlier protocols (Dudek et al, 2013). In short, precursor polypeptides were synthesized from plasmid DNA or mRNA using TNT-coupled or nuclease-treated reticulocyte lysate, respectively. In case of co-translational transport experiments, the translation reactions were supplemented with [$^{35}$S]-methionine and SPC or buffer (negative control) and incubated at 30°C for up to 60 min. Alternatively, precursor polypeptides were fully synthesized and radioactively labeled with [$^{35}$S]-methionine in reticulocyte lysate for 15 min at 30°C in the absence of SPC. After 5 min of incubation with RNase A (final concentration: 80 $\mu$g/ml) and cycloheximide (final concentration: 100 $\mu$g/ml) at 30°C, buffer or SPC were added and the reaction was continued for 30 min at 30°C (post-translational transport). All samples from transport experiments were analyzed by SDS–PAGE and phosphorimaging (Typhoon-Trio imaging system). ImageQuant TL software 7.0 was used for quantifications. Dot plots depict relative transport efficiency, which was calculated as the proportion of ER luminal precursor processing (N-glycosylation and/or signal peptide cleavage) of the total amount of synthesized precursor with the WT sample set to 100%.

### Renin activity in extracellular medium

Cultivation and transient transfection of cells stably expressing the WT, V67G, or T185A variant of *SEC61A1*–FLAG, respectively, was performed as described in section "Western blot analysis of extracellular medium." 24 h after lipofection the medium was collected, Trypsin/EDTA was added in a ratio of 1:3 and the mix was incubated for 30 min at 37°C for cleavage of the pro-region from

secreted prorenin. The reaction was stopped by adding 2.6 $\mu$l of 50 mM PMSF and incubated for 15 min at room temperature. 100 $\mu$l of medium was placed in a 96-well plate and 50 $\mu$l of the 100× diluted renin substrate conjugated with 5-FAM and QXL520 were added. Secreted renin separated the synthetic renin substrate from its quencher and the fluorescent signal was monitored at 528 nm every 15 min for 3 h at 37°C on an Infinite 2000Pro microplate reader (Tecan). Each sample was measured in triplicates.

### Colocalization study of UMOD, REN, or REN variants transiently expressed in HEK cells stably producing WT, V67G, or T185A variants of SEC61α-FLAG

$1.5 \times 10^5$ HEK293 cells stably producing WT or mutated Sec61α were grown on 1.8 cm$^2$ glass four-chamber slides (BD Falcon). After 24 h, the cells were transiently transfected by Lipofectamine TM3000 with 1 $\mu$g plasmid DNA encoding the cDNA for REN, APE-REN, REN-D2R, or UMOD according to manufacturer's protocol. 24 h after lipofection, the cells were quickly washed with PBS and fixed by cold 100% methanol for 10 min at −20°C, washed three times in PBS and proteins were blocked for 30 min at room temperature in PBS with 5% FCS. For colocalization of renin (or the renin variants) and uromodulin with the ER, cells were incubated 1 h at 37°C either with rabbit primary antibody anti-Human Preprorenin 288–317 (Yanaihara) at a dilution of 1:50 or with rabbit polyclonal primary anti-uromodulin (H-135; Santa Cruz Biotechnology) at a dilution of 1:300. Co-staining of the ER was performed with the mouse monoclonal anti-PDI antibody (Enzo Life Sciences, Inc.) at a dilution of 1:200. All antibodies were diluted in PBS with 5% FCS and 0.05% Tween-20. After incubation, the cells were washed five times in PBS and incubated 1 h at 37°C with donkey anti-rabbit Alexa Fluor 555 and goat anti-mouse Alexa Fluor 488 (Invitrogen) at a 1:1,000 dilution. Cells were washed four times and the slides were mounted in the fluorescence mounting medium ProLong Gold Antifade Mountant with DAPI before being analyzed by confocal microscopy. XYZ images were sampled according to Nyquist criterion using a Leica SP8X laser scanning confocal microscope, HC PL Apo objective (633, numerical aperture 1.40), 405 nm diode/50 mW DMOD Flexibl, and 488 and 555 laser lines in 470–670 nm 80 MHz pulse continuum WLL2. Images were restored using a classic maximum likelihood restoration algorithm in the Huygens Professional Software (SVI) (Landmann, 2002). The Pearson coefficients of the colocalization and colocalization maps using single-pixel overlap coefficient values ranging from 0 to 1 were created in the Huygens Professional Software (Manders et al, 1993). The resulting overlap coefficient values are presented as the pseudo color whose scale is shown in the corresponding lookup tables.

### Statistical analysis and graphical representation

Graphs were visualized using GraphPad Prism 5 or SigmaPlot 10 software. ANOVA in combination with Dunnett's multiple comparisons test was used for statistical comparison of multiple groups. When two groups were compared an unpaired, two-tailed *t* test was the method of choice. In the case of Western blotting and protein transport experiments, for each repeat the data set was normalized to the WT cells. *P*-values are indicated by asterisks or hashtags with

## Supplementary Information

## Acknowledgements

We are grateful to N Borgese (Milan, Italy) and S Haβdenteufel (Rehovot, Israel) for providing the plasmids encoding for cytb5-ops28 and ppl-apelin, respectively. We also like to thank S High (Manchester, UK) for providing the plasmids for MHC class II invariant chain and Sec61β-ops13 as well as T Ryan (New York, USA) and J de Juan-Sanz (Paris, France) for sharing the ER-GCamP$_{6-150}$. We acknowledge the excellent technical assistance provided by M Lerner (Homburg, Germany) and like to emphasize the invaluable input by R Zimmermann and M van der Laan. This work was supported by the Deutsche Forschungsgemeinschaft (DFG) grants SFB 894 (S Lang and A Cavalié), IRTG 1830 (S Lang), and He3875/15-1 (V Helms). A Tirincsi was supported by the HOMFORexcellent program of Saarland University Medical Center. M Živná, S Kmoch, V Barešová, J Sovová, AJ Bleyer, and P Vyleťal were supported by grant NU21-07-00033 from the Ministry of Health of the Czech Republic. M Živná, S Kmoch, V Barešová, K Hodaňová, P Vyleťal, J Sovová, and I Jedličkova were supported by institutional program UNCE/MED/007 of the Charles University in Prague.

### Author Contributions

M Sicking: conceptualization, formal analysis, investigation, visualization, methodology, and writing—original draft, review, and editing.

M Živna: conceptualization, resources, investigation, visualization, methodology, and writing—original draft, review, and editing.

P Bhadra: resources, software, methodology, and writing—original draft, review, and editing.

V Barešova: resources, formal analysis, visualization, methodology, and writing—original draft, review, and editing.

A Tirincsi: formal analysis, investigation, and writing—review and editing.

D Hadzibeganovic: formal analysis, methodology, and writing—review and editing.

KŸ Hodaňova: resources, methodology, and writing—review and editing.

P Vyleťal: resources, methodology, and writing—review and editing.

J Sovova: resources, methodology, and writing—review and editing.

I Jedličkova: resources, methodology, and writing—review and editing.

M Jung: resources, supervision, methodology, and writing—review and editing.

T Bell: conceptualization, resources, methodology, and writing—review and editing.

V Helms: conceptualization, resources, supervision, funding acquisition, methodology, and writing—original draft, review, and editing.

AJ Bleyer: conceptualization, resources, methodology, and writing—review and editing.

S Kmoch: resources, supervision, funding acquisition, methodology, and writing—review and editing.

A Cavalié: conceptualization, resources, software, supervision, funding acquisition, methodology, and writing—original draft, review, and editing.

S Lang: conceptualization, resources, supervision, funding acquisition, visualization, methodology, project administration, and writing—original draft, review, and editing.

### Conflict of Interest Statement

The authors declare that they have no conflict of interest.

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
