## [Reviewer comments · Life Science Alliance]

Life Science Alliance

Phenylbutyrate Rescues the Transport Defect of the Sec61 α Mutations V67G and T185A for Renin

Mark Sicking, Martina Živná, Pratiti Bhadra, Veronika Barešová, Andrea Tirinci, Drazena Hadzibeganovic, Kateřina Hodaňová, Petr Vyleřal, Jana Sovová, Ivana Jedličkova, Martin Jung, Thomas Bell, Volkhard Helms, Anthony Bleyer, Stanislav Kmoch, Adolfo Cavalié, and Sven Lang

DOI: <https://doi.org/10.26508/lsa.202101150>

Corresponding author(s): Sven Lang, Saarland University

Review Timeline:

Submission Date:	2021-07-07
Editorial Decision:	2021-08-04
Revision Received:	2021-12-13
Editorial Decision:	2021-12-28
Revision Received:	2022-01-06
Accepted:	2022-01-06

Transaction Report:

August 4, 2021

Re: Life Science Alliance manuscript #LSA-2021-01150-T

Dr. Sven Lang
Saarland University
Medical Biochemistry & Molecular Biology
Kirrberger Str. 100
Building 44
Homburg, Saarland 66421
GERMANY

Dear Dr. Lang,

Thank you for submitting your manuscript entitled "Phenylbutyrate rescues transport defects of the disease-associated Sec61 α mutations V67G and T185A" to Life Science Alliance. The manuscript was assessed by expert reviewers, whose comments are appended to this letter. We invite you to submit a revised manuscript addressing the Reviewer comments.

Thank you for this interesting contribution to Life Science Alliance. We are looking forward to receiving your revised manuscript.

Sincerely,

B. MANUSCRIPT ORGANIZATION AND FORMATTING:

Reviewer #1 (Comments to the Authors (Required)):

In this paper, Sicking et al investigated the effects of the SEC61 alpha mutants V67G and T185A, related with autosomal dominant tubulointerstitial kidney disease (ADTKD), on calcium homeostasis and on protein import into the ER. By using cellular models, they showed a reduced protein expression of the calcium transporters Orai1 and SERCA2 in the SEC61A1 mutants as compared to WT, and a calcium leakage from the ER lumen for the T185A mutant but not for the V67G mutant. Next, they measured in these SEC61A1 mutants a significant reduced import into the ER for the proteins REN, UMOD, but with the strongest effect on bovine PPL. The authors also demonstrated the retention of REN and UMOD in the ER lumen that could be linked with the targeting sequence. Finally, they used the molecular chaperone phenylbutyrate to revert the imbalanced calcium homeostasis seen in the SEC61A1 mutants and to restore the defective protein transport of renin.

This manuscript is clearly written, and the work is generally well executed. Overall, I found this to be a sound and worthwhile study that clearly advances the field. However, this study is still missing some critical experiments, especially to confirm their hypothesis on PBA rescue. In fact, at this stage, the title of this manuscript and their conclusion are too general, given that the authors only demonstrated the rescue with PBA of one single protein, renin. Also, the discussion is too extensive and should be rewritten more condense.

Major comments:

1. If thapsigargin treatment (ER calcium) for the V67G mutant gives the same calcium peak as the WT, how can the authors conclude that the reduced Ca^{2+} storage capacity of ADTKD-SEC61A1 cells (Fig. 2A, 2B) must originate in the ER (line 183). Here, the two mutants V67 and T185 seem to have distinct effect on calcium homeostasis at the ER, although they have the same reduced protein levels of SERCA2. And on line 204 the authors state that "the Sec61 α mutations V67G and T185A caused a reduction of the ER Ca^{2+} content", however, the effect was seen with ionomycin, which induces the release of total pan-organelle Ca^{2+} , so not specific for the ER. The interpretation of the calcium data seems not logic to this reviewer.
2. Figure 3: for REN, UMO and PPL, the calculated reduced import of these proteins seems to originate mainly from an accumulation of the precursor fraction, rather than a reduction in the mature protein fraction. Have the authors performed a quantification of these proteins in the (intact) mutant cells (like Fig 1) to verify that there is a reduced ER import and expression of REN and UMO in the 'natural' mutant cells? Additional PK treatment of the samples of Fig3A are needed to verify that the 'p' fraction is the non-translocated protein (and thus susceptible to PK degradation), and not a translocated but non-glycosylated species (that are PK resistant). If the SEC61 mutation affects the glycosylation efficiency, maybe this (not-glycosylated mature protein) can be the trigger for ER retention of REN and UMO as shown in Fig 4.
3. How do the authors correlate the reduced protein import of REN and UMO into the ER lumen (Fig 3) in the mutant cells with the enhanced accumulation (retention) of these proteins in the ER lumen? Is there not simply an accumulation of these proteins in the SEC61 channel where the precursors get blocked? In Fig 4, can the authors distinguish between a protein in the ER membrane/Sec61 channel and a protein fully translocated and present in the ER lumen? If 'retention in the ER' is a trapping of the preprotein in the ER membrane (e.g. inside translocon), that could explain the reduced protein import and reduced protein expression.
4. For pPL, there is a very clear and uniform reduction in protein import in the V67G mutant (Fig 3A and Fig 5A), but a much higher variation (going from 50 to 100%) for the T185A mutant which also explains the non-significant effect for the PPL-APE construct of Fig 5. Do the authors have an explanation for this difference of pPL translocation between the 2 mutants, given that pPL has a very strong and efficient signal peptide? Does the SP of PPL bind differently to the translocon of the SEC61 alpha mutants?
5. The effect of PBA has only been studied on the import of REN. To conclude and to state that PBA can revert the impairment of protein handling in presence of ADTKD-SEC61A1 mutations (line 339), the authors should also test the effect of PBA treatment on UMOD, and certainly on bovine PPL, given that its import is affected the most in those SEC61 mutants. Additional comparative experiments with human PPL could further support their hypothesis on the importance of early negatively charged amino acids in SP (line 442 and Fig 6C).
6. Is the effect of PBA on calcium flux (Fig 5G and H) related to a better expression of SERCA2 in the mutant cells? The authors should check the protein expression of SERCA2 in WT and ADTKD-SEC61A1 cells treated with PBA for 36 hours. Is the

translocation and membrane insertion of SERCA2 improved by PBA treatment in the mutant cells?

7. The data of Figure 6 should be described as a last paragraph of the results section. It would be interesting and relevant to include UMOD in Fig 6C, and discuss this in the discussion, especially given that an effect was seen with the V67G mutant and not the T185A mutant.

8. To reduce the length of the discussion, the hypothetical interaction section of PBA on page 16 should be removed. At this stage, this is too speculative, and there are not enough data at this stage yet to support this hypothesis. Also, the effect of PBA is only shown here for 1 substrate.

Minor comments:

- The first sentence of the second paragraph of the introduction (line 64) should be moved to the next paragraph (line 85)
- Fig 1: indicate in legend what EV stands for
- The authors should briefly comment on the reduced expression on STIM1 that is seen only in the T185 mutant and not in the V67 mutant. Can this be linked to the enhanced ER calcium leakage (Fig 2C) for the T185 mutant?
- Fig 3: list of abbreviations in the legend is missing REN and UMO
- Figure 4 C and D: the effect on Renin expression in the V67G mutant cells versus T185 mutant measured by western blot (panel D) looks stronger and more pronounced as with the fluorescence technique of panel C. Is there a loading control used for the WB (or quantification of amount of cells used for the secretion of REN). How did the authors compare the samples in terms of same amount of cells used?
- Fig 5A: Relative transport panels of REN and APE are identical to the ones used in Fig 3A and C; the authors should remove these 2 panels from Fig 5
- Uromodulin is abbreviated as UMO in the figures and as UMOD in the text; consistent use of the same abbreviation in text and figure would improve clarity of the manuscript.
- Line 389: according to fig 4B there is also ER retention of REN, which then will lead to reduced secretion, so for both UMOD and REN the main 'cause' is retention of the protein in the ER. Sentence of line 389 should be rephrased accordingly.
- PBA doses in patients: does this correlate with the rather high (3 μ M) concentration of PBA used in the experiments. Can such a dose be achieved in patients? The authors should briefly discuss this as well.

Reviewer #2 (Comments to the Authors (Required)):

The manuscript describes in detail the impact of two kidney disease-linked mutations of Sec61a with respect to cotranslational translocation and substrate selectivity, which appears to be dependent on signal peptide sequences. The polypharmacological agent phenylbutyrate was shown to attenuate defective transport of the secretory protein renin and calcium homeostasis disturbances.

The biological experiments are nicely done. My main question relates to compound screening and functionally relevant activity phenylbutyrate (PBA):

1) Only 10 compounds were screened and PBA was the (one) hit. A 10% hit rate is quite high for random screening and usually less than 0.1%. The authors should provide details about the (presumably rational) selection of the screening compounds and provide the identity/structures of the other 9 compounds that did not show activity.

2) Which biological activity is relevant to the rescue effect or is it a combination? Since PBA is also an HDAC inhibitor, in addition to being a chemical chaperone (and probably many other activities), it might be necessary to dissect the polypharmacology by assessing the phenotype upon treatment with class I or class II HDAC inhibitors, certain ER stress modulators, and other selective agents. Maybe they were included in the 10-membered library? Clarification is needed here. Overall, the functional link should be strengthened.

Otherwise, I have no serious concerns. However, the Discussion section appears to be unusually long and a more concise discussion of the data would be beneficial.

Reviewer #1 (Comments to the Authors (Required)):

In this paper, Sicking et al investigated the effects of the SEC61 alpha mutants V67G and T185A, related with autosomal dominant tubulointerstitial kidney disease (ADTKD), on calcium homeostasis and on protein import into the ER. By using cellular models, they showed a reduced protein expression of the calcium transporters Orai1 and SERCA2 in the SEC61A1 mutants as compared to WT, and a calcium leakage from the ER lumen for the T185A mutant but not for the V67G mutant. Next, they measured in these SEC61A mutants a significant reduced import into the ER for the proteins REN, UMOD, but with the strongest effect on bovine PPL. The authors also demonstrated the retention of REN and UMOD in the ER lumen that could be linked with the targeting sequence. Finally, they used the molecular chaperone phenylbutyrate to revert the imbalanced calcium homeostasis seen in the SEC61A1 mutants and to restore the defective protein transport of renin. This manuscript is clearly written, and the work is generally well executed. Overall, I found this to be a sound and worthwhile study that clearly advances the field. However, this study is still missing some critical experiments, especially to confirm their hypothesis on PBA rescue. In fact, at this stage, the title of this manuscript and their conclusion are too general, given that the authors only demonstrated the rescue with PBA of one single protein, renin. Also, the discussion is too extensive and should be rewritten more condense.

We very much appreciate the time and effort taken by reviewer #1 to critically assess the data we presented and providing detailed as well as stimulating feedback. As requested, we have rephrased and toned the title as well as shortened the discussion. The new title reads as follows: "Phenylbutyrate Rescues the Transport Defect of the Sec61 α Mutations V67G and T185A for Renin".

Major comments:

1. If thapsigargin treatment (ER calcium) for the V67G mutant gives the same calcium peak as the WT, how can the authors conclude that the reduced Ca²⁺ storage capacity of ADTKD-SEC61A1 cells (Fig. 2A, 2B) must originate in the ER (line 183). Here, the two mutants V67 and T185 seem to have distinct effect on calcium homeostasis at the ER, although they have the same reduced protein levels of SERCA2. And on line 204 the authors state that "the Sec61 α mutations V67G and T185A caused a reduction of the ER Ca²⁺ content", however, the effect was seen with ionomycin, which induces the release of total pan-organellar Ca²⁺, so not specific for the ER. The interpretation of the calcium data seems not logic to this reviewer.

We have slightly rephrased the corresponding sentences to improve the interpretation of the line of experiments and the underlying logic. Important for the conclusion that the reduced calcium storage capacity originates in the ER are the data presented in Fig. 2E and 2F. After specific depletion of the ER calcium store via thapsigargin (see peaks at ~200 s in Fig. 2E) the subsequent application of ionomycin reveals the remaining calcium storage capacity of compartments other than the ER (see peaks at ~1000 s in Fig. 2E). These latter peaks are identical for the wild type and mutant carrying cell lines and can be interpreted that calcium storage in the other organelles are identical for the cell lines. Thus, the difference of the total cellular calcium pool found in Fig. 2A and 2B, i.e., by the direct application of ionomycin, most likely originates in the ER. Important to note, and as stated in the text, these measurements were performed in the absence of extracellular calcium. Hence, only

intracellular calcium stores were monitored in the course of the measurements. The absence of extracellular calcium avoids any refilling of the internal stores during the measurement.

The updated text passage reads as follows: “Furthermore, the consecutive application of thapsigargin and ionomycin in a setting devoid of extracellular Ca²⁺ helped us to demonstrate that after full depletion of the ER Ca²⁺ store (achieved by thapsigargin) the remaining Ca²⁺ storage capacity of organelles other than the ER (revealed by ionomycin) is identical for the control and mutant cells (Fig. 2E, 2F). In other words, the reduced Ca²⁺ storage capacity of ADTKD-SEC61A1 cells (Fig. 2A, 2B) most likely originates in the ER (Fig. 2E, 2F) and is probably linked to the reduced abundance of SERCA2 (Fig. 1).”

2. Figure 3: for REN, UMO and PPL, the calculated reduced import of these proteins seems to originate mainly from an accumulation of the precursor fraction, rather than a reduction in the mature protein fraction. Have the authors performed a quantification of these proteins in the (intact) mutant cells (like Fig 1) to verify that there is a reduced ER import and expression of REN and UMO in the 'natural' mutant cells? Additional PK treatment of the samples of Fig3A are needed to verify that the 'p' fraction is the non-translocated protein (and thus susceptible to PK degradation), and not a translocated but non-glycosylated species (that are PK resistant). If the SEC61 mutation affects the glycosylation efficiency, maybe this (not-glycosylated mature protein) can be the trigger for ER retention of REN and UMO as shown in Fig 4.

The reviewer raises an interesting point, that an impact on glycosylation efficiency can be the trigger for ER retention and accumulation of renin and uromodulin. Based on the Western blot results for the oligosaccharyltransferase (OST) complex subunit ribophorin 1 (RPN1; Fig. 1A), a reduced abundance of the OST complex affecting glycosylation in general seems not to be the case. Along the same line, the set of substrates analyzed in addition to renin, uromodulin, and preprolactin rather excludes a general impairment of glycosylation efficiency and protein processing in the ER of ADTKD cells. For example, the Sec61-independently transported substrates cytochrome b5 and Sec61β (Lang et al. 2012; doi:10.1242/jcs.096727) showed no defect of protein glycosylation in the mutant ER fractions (CYT, 61B; Fig. 3 C, 3D). Similarly, the short precursor protein apelin (APE; Fig. 3C, 3D) and the single-spanning ER membrane protein invariant chain (IVC; Fig. 3A, 3B) that are inserted into the ER in a Sec61 complex dependent fashion also showed no impairment of glycosylation. Based on the substrates preprocecropin (CEC; Fig. 3C, 3D) as well as prion protein (PRP; Fig. 3A, 3B) a defect in signal peptide cleavage and, eventually, GPI-anchor attachment are rather unlikely events in the ADTKD cells. Further support for the interpretation of a signal peptide-specific impairment of protein transport is provided in Figure S4. Glycosylation of renin and uromodulin was measured in a linear range of the ER (Fig. S4A-D) and single point mutations of renin (REN-W10R) and uromodulin (UMO-C32Y) did show the same transport defect as their wild type counterparts (cf. Figs. S4E and 3A).

The 'p' fraction is indeed the non-translocated protein and fully susceptible to PK degradation as is shown in the following pictures for the affected substrates preprolactin and renin (Fig. i). The increased amount of precursor we often observed most likely represents the fraction of precursor proteins that are not transported and are not yet degraded in the reticulocyte lysate.

[Figure removed by editorial staff per authors' request]

We share the opinion of the reviewer and it would be ideal to quantify the level of endogenous renin and uromodulin protein in the different cell lines. Yet, as secreted proteins the intracellular levels of renin and uromodulin in intact cells is assumed to be low. Alternatively, we tested for the expression of the renin and uromodulin gene and subjected the four cell lines to RNAseq analysis. Based on the FPKM values as a measure of gene expression, this analysis showed almost no expression of renin in the HEK lines and the expression of uromodulin could not be detected at all (Tab. 1). This also explains why we used transient expression of renin and uromodulin for their identification and detection in figure 4. For comparison, the FPKM values for the *ATP2A2* gene (encoding for the SERCA2 protein) are given (s. also below answer 6).

Table 1: Expression values (FPKM; Fragments per kilobase per million mapped reads)

identified by RNAseq analysis using the ADTKD cell models			
Gene of interest	Cell line	FPKM values (RNAseq)	Average FPKM
RENIN	EV	0.0954; 0.1773; 0.0824	0.1184
	WT	0; 0.0254; 0.0272	0.0175
	V67G	0.0265; 0.0660	0.0463
	T185A	0; 0; 0	0
UROMODULIN	EV	Not identified	-
	WT	Not identified	-
	V67G	Not identified	-
	T185A	Not identified	-
ATP2A2	EV	130.22; 107.91; 114.77	117.63
	WT	144.16; 102.70; 114.35	120.40
	V67G	124.02; 107.82	115.92
	T185A	112.71; 115.14; 127.01	118.29

3. How do the authors correlate the reduced protein import of REN and UMO into the ER lumen (Fig 3) in the mutant cells with the enhanced accumulation (retention) of these proteins in the ER lumen? Is there not simply an accumulation of these proteins in the SEC61 channel where the precursors get blocked? In Fig 4, can the authors distinguish between a protein in the ER membrane/Sec61 channel and a protein fully translocated and present in the ER lumen? If 'retention in the ER' is a trapping of the preprotein in the ER membrane (e.g. inside translocon), that could explain the reduced protein import and reduced protein expression.

As is suggested by the reviewer, we also think that the accumulation of renin and uromodulin might occur at the level of the mutant SEC61 channel and thereby, at least temporarily, blocking it. Instead of a permanent blockage, certain signal peptides (like the one from renin) might not trigger opening of the mutant Sec61 complexes efficiently and thereby increasing the dwell time of such a substrate. Regarding the protein transport assay used in Figure 3, those scenarios of permanent/transient blockage or inefficient gating of mutant Sec61 complexes can account for the partially reduced transport activity we observed. In addition, we speculated if the Sec61 mutations might affect proper modification and folding of renin and uromodulin in the ER lumen. However, such an effect might be secondary in nature. For example, either the abundance of other ER proteins that are required for efficient folding or modification could be reduced, or their activity is hampered by the lower calcium level in the ER of the mutant cells (Fig. 2). As shown in Fig. 1, the abundance of classic ER chaperones (BiP, Grp170, PDI) is not affected. Furthermore, based on the efficient transport of the BiP-dependent substrate apelin (APE, Fig. 3C; Haßdenteufel et al. 2018; <https://doi.org/10.1016/j.celrep.2018.03.122>) the activity of BiP seems to be sufficient in the mutant ADTKD cells. Overall, we see the reduced transport of certain substrates that eventually is followed by insufficient maturation and retention in the ER.

Figure 4 shows the colocalization of renin and uromodulin with a marker of the ER lumen (PDI) that is not changed in its abundance (PDI; Fig. 1). The wild type and mutant HEK lines were transiently transfected by plasmids encoding for uromodulin (Fig. 4A) or renin (Fig. 4B). The resolution of fluorescence microscopy that we have used does not allow for the precise localization of the proteins in or at the ER. Confocal microscopes may reach resolutions of 180 nm laterally and 500 nm axially, so we are not able to distinguish between the proteins located in the lumen of the ER and proteins associated with the ER membrane as was discussed above for the precursor proteins accumulating in or at the Sec61 complex. One idea is, that future experiments will be directed at the colocalization of renin with the FLAG-tagged wild type or mutant Sec61 α proteins using super-resolution microscopy. Such experiments might be able to demonstrate that a precursor like renin indeed accumulates at the mutant Sec61 complex.

4. For pPL, there is a very clear and uniform reduction in protein import in the V67G mutant (Fig 3A and Fig 5A), but a much higher variation (going from 50 to 100%) for the T185A mutant which also explains the non-significant effect for the PPL-APE construct of Fig 5. Do the authors have an explanation for this difference of pPL translocation between the 2 mutants, given that pPL has a very strong and efficient signal peptide? Does the SP of PPL bind differently to the translocon of the SEC61 alpha mutants?

The reviewer correctly observed that the T185A mutant shows a higher variability for the transport of preprolactin. Although it was not a general phenomenon, other substrates like renin, invariant

chain, or apelin also showed a higher variability in their transport efficiency in presence of the T185A mutant cells. From a mechanistic point of view, we are unable to provide a definitive explanation for the difference in variability. As shown in Figure 6A and 6B, the V67 and T185 residues are located in interesting positions within the Sec61 complex and the corresponding mutations (V67G, T186A) might interfere differently with the Sec61 mediated protein transport and calcium leakage. The V67 amino acid sits at the tip of the plug domain in the closed conformation of the Sec61 complex. During opening of the Sec61 complex by a signal peptide and its “transitory integration” into the lateral gate the signal peptide might come in close contact with the V67 residue or the plug domain. Interestingly, V67 is flanked by the positively charged R66. In the context of the V67G mutation, the smaller side chain and helix breaking potential of glycine might increase the accessibility of a signal peptide to the R66 residue. Thereby, the negative charge in the signal peptides of renin and preprolactin could cause an unproductive interaction (D2-R66) that slows efficient opening of the Sec61 complex carrying the V67G mutation. On the other hand, in the wild type Sec61 α protein the residue T185 sits in the center of transmembrane helix 5. In the closed conformation, T185 faces away from the central pore but rotate towards it during opening of the Sec61 complex. Intriguingly, in this open conformation T185 lies opposite to the lateral gate and is part of the tunnel wall that shapes the pore (Fig. 6B). Thus, during the translocation of a nascent chain T185 might come in contact with the polypeptide in transit. Therefore, the T185A mutation could interfere with the translocation process of certain polypeptides directly or interfere with the structural mobility of the Sec61 complex. The loss of the hydroxyl-group due to the exchange of threonine to alanine also impacts the calcium leak, an effect specifically observed for the T185A mutant (Fig. 2D).

Lastly, to address the question “Does the SP of PPL bind differently to the translocon of the SEC61 alpha mutants?” more directly, we attempted chemical crosslink experiments (Fig. ii). As seen in the autoradiogram below, no profound difference in the crosslink pattern or crosslink efficiency between the wild type and mutant Sec61 α proteins with the truncated ppl86 variant could be detected. Keeping in mind the heterozygosity of the cell lines that produce in addition to the endogenous Sec61 α protein also the FLAG-tagged wild type or mutant Sec61 α such crosslink data become difficult to interpret. Despite the small tag, we were unable to differentiate if the arrested nascent chain (ppl86) crosslinked to the endogenous, untagged Sec61 α protein and/or to the FLAG-tagged Sec61 α variants.

[Figure removed by editorial staff per authors' request]

5. The effect of PBA has only been studied on the import of REN. To conclude and to state that PBA can revert the impairment of protein handling in presence of ADTKD-SEC61A1 mutations (line 339), the authors should also test the effect of PBA treatment on UMOD, and certainly on bovine PPL, given that its import is affected the most in those SEC61 mutants. Additional comparative experiments with human PPL could further support their hypothesis on the importance of early negatively charged amino acids in SP (line 442 and Fig 6C).

As before, we agree with the reviewer and her/his detailed insights and critique. To avoid any overselling of the data, we have adjusted the title of the manuscript and adjusted the statement in line 339 to acknowledge the limited set of substrates tested with PBA. The sentence in question was changed to:

“This conclusion was further supported by the identification of PBA as small molecule that can reverse the impairment of renin secretion and Ca²⁺ handling in the presence of ADTKD-SEC61A1 mutations.”

In general, to study the effect of PBA in more detail we are currently developing a heterozygous mouse model to recapitulate the disease model in a mammalian organism and test the impact of a PBA treatment on multiple physiological parameters including renin and uromodulin secretion by the kidney. So far, our best readout is for renin, which showed improved transport (Fig. 4C) and secretion

(Fig. 4E) in presence of PBA. Indeed, we tried using the bovine ppl in combination with PBA, however, the data were inconclusive as PBA seemed to preferentially reduce the ppl transport in the wildtype cells giving the false impression of a rescue. From a medical perspective an improved transport for preprolactin, bovine or human, would most likely not contribute to any betterment of the kidney disease phenotype. In our hands the human preprolactin isn't a well-suited transport substrate mainly for two reasons. One, it's overall transport efficiency is lower compared to that of the bovine homolog. Two, in contrast to the bovine ppl the human ppl carries a glycosylation site. Thus, upon entering the ER the cleavage of the signal peptide and addition of the N-glycan prevent an easy readout of the transport without subsequent sequestration or PNGase/EndoH treatment of the reaction. Therefore, we decided to test the hypothesis of the importance of early negatively charged amino acids in the signal peptide by a renin variant, REN-D2R. In the REN-D2R variant the negatively charged aspartate in position 2 was replaced by an arginine. REN-D2R was tested in two assays, the in vitro transport (Fig. iii) and immunofluorescence colocalization (Fig. iv). In both assay the charge reversal suppressed the issues with protein transport and accumulation at or in the ER. The data for REN-D2R were added to the manuscript in Fig. S4E and Fig. S8.

Figure iii: Protein transport of the renin variant REN-D2R in presence of ADTKD-*SEC61A1* mutations

The radioactively labeled precursor polypeptides renin-D2R (REN-D2R), whose aspartate in position two of the signal peptide is substituted by an arginine, was imported into the indicated ER fractions under co-translational transport conditions. Reactions devoid of any ER fraction served as negative control (no ER). ER fraction refers to semi-permeabilized cells generated from the empty vector (EV) or the heterozygous wild type (WT), V67G, and T185A cell lines by digitonin treatment and S7 nuclease mediated mRNA degradation. In contrast to the reduced transport observed for renin in presence of V67G or T185A mutation, the REN-D2R variant shows similar transport rates into the WT, V67G, and T18A ER fractions. m, matured polypeptide localized in the ER; p, precursor polypeptide.

Figure iv: Colocalization studies of the renin variant REN-D2R in presence of ADTKD-*SEC61A1* mutations
 Immunofluorescence images of wild type (WT) and the two ADTKD-*SEC61A1* cell types were taken 24 h after transient transfection with an expression plasmid encoding for the renin variant REN-D2R. Immunostainings show the localization of REN-D2R and an ER marker protein, protein disulfide isomerase (PDI). The degree of colocalization is shown by the overlay (REN-D2R, red; PDI, green; nucleus, blue) and colocalization images (last column). The overlap of recorded fluorescent signals was quantified by the pseudo-colored overlap coefficient that ranges from 0 to 1 and is given as separate scale at the bottom of the figure. A scale bar (5 μ m) for each row of pictures is provided. Pearson's correlation coefficients \pm SEM between red and green signals are displayed in the colocalization images as white numbers. In contrast to the more pronounced colocalization observed between wild type renin and PDI in presence of V67G or T185A mutation, the REN-D2R variant shows a similarly low degree of colocalization in the WT, V67G, and T18A cells.

As suggested further, we did test the effect of PBA on the transport of uromodulin (Fig. v). No improvement of the hampered uromodulin transport in presence of the V67G mutations could be detected. As stated in the discussion, PBA did not improve the transport or secretion of mutant uromodulin (ADTKD-*UMOD*) in mouse models either (Kemter et al., 2014).

[Figure removed by editorial staff per authors' request]

6. Is the effect of PBA on calcium flux (Fig 5G and H) related to a better expression of SERCA2 in the mutant cells? The authors should check the protein expression of SERCA2 in WT and ADTKD-SEC61A1 cells treated with PBA for 36 hours. Is the translocation and membrane insertion of SERCA2 improved by PBA treatment in the mutant cells?

The reviewer points out a potential mechanism that could help to explain our finding of PBA restoring the cellular Ca^{2+} content (Fig. 5F-I). We tried to compare SERCA2 expression in the different cell lines without and with PBA treatment after 36 hours using qRT-PCR (Fig. vi). The bar graph below shows the relative SERCA2 mRNA levels. Expression of the SERCA2 gene (*ATP2A2*) was neither different in the WT and mutant cell lines nor was it induced by the application of 0.5 mM PBA. In addition, using the same RNAseq data set mentioned above for renin and uromodulin no significantly altered expression of the *ATP2A2* gene could be detected (s. Tab. 1). The comparative log2 fold changes for the V67G vs WT and T185A vs WT were -0.095 (= 94%) and -0.028 (= 98%), respectively. In both cases the adjusted p-values were calculated as 0.95, indicating no significant change in expression.

[Figure removed by editorial staff per authors' request]

A similar finding was previously reported by Takada *et al.* 2012 (<https://doi.org/10.1371/journal.pone.0039893>). The authors compared SERCA2 expression in heart tissue from different rat models without and with PBA treatment. PBA did not provide a significant difference in SERCA2 gene expression. Regarding the SERCA2 protein content, the authors found opposite effects for PBA depending on the rat model they used. The PBA treatment slightly decreased the SERCA protein level in one model (LETO), but increased it in a second model (OLETF). Unfortunately, our Western blot analysis for the SERCA2 protein level upon PBA treatment remained inconclusive as of yet. After re-ordering the commercial SERCA2 antibody (same company, but a different batch) we were sadly not able to detect the protein in the same quality as shown in Fig. 1. We apologize for this circumstance and hope to find an alternative (e.g., global proteomic analysis +/- PBA) to address this interesting question in the near future more accurately. Due to the lack of a signal sequence and N-linked glycosylation we did not test SERCA2 as a substrate in the established in vitro assay. While carbonate extractions could be performed to test the stability of membrane

insertion of membrane proteins, we are uncertain if this type of assay would be sensitive enough to detect any beneficial impact of PBA on the membrane insertion of SERCA.

7. The data of Figure 6 should be described as a last paragraph of the results section. It would be interesting and relevant to include UMOD in Fig 6C, and discuss this in the discussion, especially given that an effect was seen with the V67G mutant and not the T185A mutant.

We have done so accordingly and added a new paragraph to the results section “Structure-function relationship of the ADTKD-SEC61A1 mutations V67G and T185A”.

8. To reduce the length of the discussion, the hypothetical interaction section of PBA on page 16 should be removed. At this stage, this is too speculative, and there are not enough data at this stage yet to support this hypothesis. Also, the effect of PBA is only shown here for 1 substrate.

We have deleted the paragraph and the corresponding supplementary figure S7 as suggested.

Minor comments:

- The first sentence of the second paragraph of the introduction (line 64) should be moved to the next paragraph (line 85)

The change was made as suggested.

- Fig 1: indicate in legend what EV stands for

We added the explanation for EV to the legend of Fig. 1.

- The authors should briefly comment on the reduced expression on STIM1 that is seen only in the T185 mutant and not in the V67 mutant. Can this be linked to the enhanced ER calcium leakage (Fig 2C) for the T185 mutant?

We added a short comment about the reduced STIM1 protein level in the results section. STIM1 is the ER membrane protein that senses the reduction of the ER luminal calcium concentration and then activates the Orai channels in the plasma membrane to allow the influx of calcium. This mechanism is called the store-operated calcium entry (SOCE). We consider it rather unlikely that STIM1 is directly linked to the reduced ER calcium leakage we observed for the T185A mutant. Despite a lower abundance of STIM1 and Orai1, the remaining levels seem to suffice to trigger an efficient SOCE in the T185A and V67G cell lines (Fig. 2G).

- Fig 3: list of abbreviations in the legend is missing REN and UMO

We added the explanation for abbreviations.

- Figure 4 C and D: the effect on Renin expression in the V67G mutant cells versus T185 mutant measured by western blot (panel D) looks stronger and more pronounced as with the fluorescence technique of panel C. Is there a loading control used for the WB (or quantification of amount of cells used for the secretion of REN). How did the authors compare the samples in terms of same amount of cells used?

Both assays do show the same trend for a reduced secretion of renin. It is important to note, that two different methods were used that do not report on the exact same protein. Due to addition of the trypsin/EDTA before the fluorescent measurement in Fig. 4C, the activity of secreted renin was measured in the extracellular medium. However, the Western blot reports on the secretion of the secreted precursor form of renin called prorenin. Aside from minor fluctuations that reflect on biological variance the comparison of two different readouts and protein forms might account for the impression of a stronger effect in the Western blot. The volume of culture media used in the reaction was normalized to the number of cells at the time of medium collection (Fig. 4C). Samples for Western blot were normalized to total protein amount (Fig. 4D).

- Fig 5A: Relative transport panels of REN and APE are identical to the ones used in Fig 3A and C; the authors should remove these 2 panels from Fig 5.

To make it easier for the reader and to allow a direct comparison we kept the data displayed twice. We added a statement to the figure legend of Fig. 5 that the data set was shown before in Fig. 3.

- Uromodulin is abbreviated as UMO in the figures and as UMOD in the text; consistent use of the same abbreviation in text and figure would improve clarity of the manuscript.

We have changed the abbreviation to UMOD throughout the manuscript text and figures.

- Line 389: according to fig 4B there is also ER retention of REN, which then will lead to reduced secretion, so for both UMOD and REN the main 'cause' is retention of the protein in the ER. Sentence of line 389 should be rephrased accordingly.

The sentence was rephrased accordingly to "Thus, the manifestation of a kidney disease pattern in patients carrying the Sec61 α -V67G or Sec61 α -T185A mutation could very well be explained by the substrate-specific retention of REN and eventually UMOD in or at the ER, thereby hampering their efficient secretion."

- PBA doses in patients: does this correlate with the rather high (3 μ M) concentration of PBA used in the experiments. Can such a dose be achieved in patients? The authors should briefly discuss this as well.

We have added some information about the dosage of PBA approved for the treatment of patients.

The recommendation by FDA for the dosing of PBA (called Buphenyl) is given with 450-600 mg/kg/day in patients weighing less than 20 kg, or 9.9-13.0 g/m²/day in larger patients. The drug should be taken in equally divided amounts with each meal (i.e., three to six times per day).

- Considering an average patient with a height of 175 cm and 75 kg (5' 9" and 165 pounds), this person displays a body surface area (BSA) of 1.9 m² [using the Mosteller equation: $BSA = (\text{height (cm)} \times \text{weight (kg)} / 3600)^{0.5}$].

- Thus, the recommended dose of PBA for the average patient would be 1.9 m² x 11.0 g/m²/day \approx 21 g/day.

- Taking into account the molecular weight of PBA with 186 g/mol the daily dosage of 21 g represents \approx 110 mmol.

- As the density of the human body is roughly 1 kg/m³, the average patient provides a volume of 75 m³ or 75 l.

- In terms of concentration 110 mmol/75 liter \approx 1.5 mM.

- We are aware that this very simplified calculation ignores most principles of pharmacology regarding the LADME (liberation, absorption, distribution, metabolism, excretion) principle.

While a concentration of 1 mM might appear high, the PBA treatment is often well tolerated by patients despite the high dosage taken daily. Previously PBA was used in mice with 400 mg/kg (injection) or 60 mM in drinking water ad libitum (Jara et al. 2018, <https://doi.org/10.1016/j.exer.2018.06.015>). In another study, PBA was dissolved in drinking water and administered at a daily dosage of 1 g/kg body weight (Kemter et al. 2014, <https://doi.org/10.1074/jbc.M113.537035>).

Reviewer #2 (Comments to the Authors (Required)):

The manuscript describes in detail the impact of two kidney disease-linked mutations of Sec61a with respect to cotranslational translocation and substrate selectivity, which appears to be dependent on signal peptide sequences. The polypharmacological agent phenylbutyrate was shown to attenuate defective transport of the secretory protein renin and calcium homeostasis disturbances.

The biological experiments are nicely done. My main question relates to compound screening and functionally relevant activity phenylbutyrate (PBA):

We thank reviewer #2 for providing constructive criticism and feedback. As discussed below, we tried to answer the issues that have been raised.

1) Only 10 compounds were screened and PBA was the (one) hit. A 10% hit rate is quite high for random screening and usually less than 0.1%. The authors should provide details about the (presumably rational) selection of the screening compounds and provide the identity/structures of the other 9 compounds that did not show activity.

The reviewer is correct that a hit rate of 10 % during an unbiased screen would be rather high. However, considering the limited screening ability of the real-time cell analyzer (a single 96 well plate is scanned for multiple days) we did not intend to perform an unbiased screen. Instead, and as correctly assumed by the reviewer, we picked ten compounds to address three rationales. First, we wondered if ER stress would be involved in the pathogenesis, which was recently published for the Sec61 α -Q92R mutant (Van Nieuwenhove et al. 2020, <https://doi.org/10.1016/j.jaci.2020.03.034>). We considered multiple options. The presence of the V67G or T185A mutation could endow cells with an increased resilience or susceptibility towards inducers (tunicamycin and dithiothreitol) or suppressors (sodium phenylbutyrate, tauroursodeoxycholic acid) of ER stress. Second, we speculated if activation (ellagic acid, forskolin) or inhibition (thapsigargin) of SERCA2 activity might be affecting cell survival differently in the wild type and mutant cell lines. Third, we compared survival in presence of inhibitors of the Sec61 complex (eeyarestatin 1), the ribosome (puromycin), or the mitochondrial ATP synthase (oligomycin A). The data showed that PBA was the compound that managed to improve cell survival at two different concentrations. Of note, the effect was not specific for the mutant carrying cells and the wildtype cells benefitted from the PBA treatment, too (Fig. S5).

As requested, we added the rational for compound selection and a new supplementary figure (Fig. S6) with the structures of the compounds (cf. Fig. vii) to the manuscript.

Compound (Abbreviation) Biological activity	Structure	Compound (Abbreviation) Biological activity	Structure
Dithiothreitol (DTT) Induction of ER stress		Phenylbutyrate (PBA) Suppression of ER stress	Eeyarestatin 1 (ES1) Inhibition of Sec61 complex		Puromycin (PUR) Inhibition of protein synthesis	Ellagic acid (ELL) Activation of SERCA2		Tauroursodeoxycholic acid (TUDCA) Suppression of ER stress	Forskolin (FOR) Activation of SERCA2		Thapsigargin (TG) Inhibition of SERCA2	Oligomycin A (OLI) Inhibition of ATP synthase		Tunicamycin (TUN) Induction of ER stress	
Figure vii: Compounds and their structures used for the small-scale screening based on cell proliferation

2) Which biological activity is relevant to the rescue effect or is it a combination? Since PBA is also an HDAC inhibitor, in addition to being a chemical chaperone (and probably many other activities), it might be necessary to dissect the polypharmacology by assessing the phenotype upon treatment with class I or class II HDAC inhibitors, certain ER stress modulators, and other selective agents. Maybe they were included in the 10-membered library? Clarification is needed here. Overall, the functional link should be strengthened.

The reviewer asks a very relevant, hard to answer question. Disentangling the polypharmacology of a small molecule like PBA is rather complex and beyond the scope of our current manuscript. With the exception of other HDAC inhibitors, the small-scale screen did actually encompass additional ER stress modulators (including tunicamycin, dithiothreitol, thapsigargin, eeyarestatin 1, tauroursodeoxycholic acid) and other selective compounds (puromycin, oligomycin A). Yet, PBA emerged as the best candidate improving cell proliferation. Based on the additional data presented, multiple lines of evidence pinpoint at the molecular chaperone activity of PBA being at work rather than its inhibitory effect on HDACs. For example, the functional analysis of the Sec61 activity in the presence of PBA showed improved transport of renin (Fig. 5C) as well as an increased calcium efflux (Fig. S7). In both cases, the PBA effect happened within minutes speaking in favor of a molecular chaperone activity rather than changes in gene expression that would have to manifest at the protein level first. Furthermore, the reconstituted in vitro transport experiments (Fig. 5A-D, Fig. 3) are performed in semi-permeabilized cells, which also excludes changes in gene expression. In Fig. 5E, intact cells were transiently transfected with the renin expressing plasmid and treated with PBA for 24 hours. Here, PBA also showed a positive impact on renin secretion. This time window and experimental set up might indeed suffice to execute an HDAC inhibitor function. However, the renin plasmid does not encode for the endogenous renin promoter but uses the CMV promoter. The latter is (most likely) not regulated by acetylation and the HDACi activity of PBA. Even in this case, we interpret the rescue of renin secretion in light of the chemical chaperone activity of PBA. PBA also restored the calcium content of the ER (Fig. 5F-I) without triggering increased expression of the SERCA2 encoding gene *ATP2A2*, which was determined by qRT-PCR (s. above question 6). One idea, as suggested by reviewer 1, could be the increased protein abundance of SERCA2 protein whose membrane insertion rather than gene expression might be improved by PBA. Except for *ATP2A2*, we have not performed global transcriptomics to determine changes in the expression landscape that would further support the HDAC inhibitory function of PBA. However, this is something we will consider for the near future. One reason for PBA being considered an attractive HDAC inhibitor is its potent anti-tumor effect in vitro, that triggers growth retardation of various human cancer cells by affecting expression of oncogenes and tumor suppressor genes. Yet, the effect we observed for the stable HEK293 cell lines, which are immortalized but not malignant, was rather the opposite, an improved cell proliferation. Thus, many of our functional data speak in favor of the molecular chaperone function of PBA. Nevertheless, we are fully aware that the totality of our data were gathered, by definition, in vitro. Future experiments, in particular using a mouse model we are currently developing, will address the pharmacodynamics and -kinetics of PBA with regard to ADTKD-*SEC61A1* in vivo in more detail. Pharmacokinetics will become an important aspect of such experiments as PBA can be converted in vivo into phenylacetate via cellular β -oxidation. This metabolization of PBA to phenylacetate is an important requirement for its additional function as ammonium scavenger during the treatment of patients suffering from urea cycle disorders.

Otherwise, I have no serious concerns. However, the Discussion section appears to be unusually long and a more concise discussion of the data would be beneficial.

We fully agree and have substantially shortened the discussion.

December 28, 2021

RE: Life Science Alliance Manuscript #LSA-2021-01150-TR

Dr. Sven Lang
Saarland University
Medical Biochemistry & Molecular Biology
Kirrberger Str. 100
Building 44
Homburg, Saarland 66421
Germany

Dear Dr. Lang,

Thank you for submitting your revised manuscript entitled "Phenylbutyrate Rescues the Transport Defect of the Sec61 α Mutations V67G and T185A for Renin". We would be happy to publish your paper in Life Science Alliance pending final revisions necessary to meet our formatting guidelines.

- please add the Twitter handle of your host institute/organization as well as your own or/and one of the authors in our system
- please use the [10 author names, et al.] format in your references (i.e. limit the author names to the first 10)
- we encourage you to revise the figure legends for figures 2, 5, S3 such that the figure panels are introduced in alphabetical order
- please add callouts for Figures S1B, C; S2A-B; S4F, G; S5B-E to your main manuscript text

A. FINAL FILES:

B. MANUSCRIPT ORGANIZATION AND FORMATTING:

Sincerely,

Reviewer #1 (Comments to the Authors (Required)):

The authors did a great job in revising the manuscript substantially to address my comments. They also performed several additional experiments and provided extra data in the point-to-point reply to explain some experimental data in more detail. This manuscript is now ready for publication in Life Science Alliance!

Reviewer #2 (Comments to the Authors (Required)):

The authors have carefully and reasonably addressed all comments and concerns. I recommend publication.

January 6, 2022

RE: Life Science Alliance Manuscript #LSA-2021-01150-TRR

Dr. Sven Lang
Saarland University
Medical Biochemistry & Molecular Biology
Kirrberger Str. 100
Building 44
Homburg, Saarland 66421
Germany

Dear Dr. Lang,

Thank you for submitting your Research Article entitled "Phenylbutyrate Rescues the Transport Defect of the Sec61 α Mutations V67G and T185A for Renin". It is a pleasure to let you know that your manuscript is now accepted for publication in Life Science Alliance. Congratulations on this interesting work.

DISTRIBUTION OF MATERIALS:

Again, congratulations on a very nice paper. I hope you found the review process to be constructive and are pleased with how the manuscript was handled editorially. We look forward to future exciting submissions from your lab.

Sincerely,
